# Seed longevity is controlled by metacaspases

Chen Liu [1,2,3,4,18], Ioannis H. Hatzianestis [2,3,18], Thorsten Pfirrmann [5], Salim H. Reza[6], Elena A. Minina [7], Ali Moazzami[7], Simon Stael[7,8,9], Emilio Gutierrez–Beltran[10,11], Eugenia Pitsili[8,9], Peter Dörmann [12], Sabine D'Andrea [13], Kris Gevaert[14,15], Francisco Romero–Campero [10], Pingtao Ding [16], Moritz K. Nowack[8,9], Frank Van Breusegem[8,9], Jonathan D. G. Jones [17], Peter V. Bozhkov [7] & Panagiotis N. Moschou [2,3,7] ✉

To survive extreme desiccation, seeds enter a period of quiescence that can last millennia. Seed quiescence involves the accumulation of protective storage proteins and lipids through unknown adjustments in protein homeostasis (proteostasis). Here, we show that mutation of all six type–II metacaspase (MCA–II) proteases in *Arabidopsis thaliana* disturbs proteostasis in seeds. MCA–II mutant seeds fail to restrict the AAA ATPase CELL DIVISION CYCLE 48 (CDC48) at the endoplasmic reticulum to discard misfolded proteins, compromising seed storability. Endoplasmic reticulum (ER) localization of CDC48 relies on the MCA–IIs-dependent cleavage of PUX10 (ubiquitination regulatory X domain–containing 10), the adaptor protein responsible for titrating CDC48 to lipid droplets. PUX10 cleavage enables the shuttling of CDC48 between lipid droplets and the ER, providing an important regulatory mechanism sustaining spatiotemporal proteolysis, lipid droplet dynamics, and protein homeostasis. In turn, the removal of the PUX10 adaptor in MCA–II mutant seeds partially restores proteostasis, CDC48 localization, and lipid droplet dynamics prolonging seed lifespan. Taken together, we uncover a proteolytic module conferring seed longevity.

The desiccation–associated phytohormone abscisic acid (ABA) promotes seed quiescence by repurposing stress–related networks into a seed dormancy program. This network rewiring also allows for storing large amounts of proteins in the developing seed to protect and nourish the embryo[1]. Given that these proteins are highly structurally disordered, it is expected that they may activate the unfolded protein response (UPR)[2]. The UPR is a proteostatic mechanism reliant on proteasomal degradation. Overall, UPR can slow down translation to help remove misfolded or even disordered proteins that cause endoplasmic reticulum (ER) stress. For this function, UPR deploys the INOSITOL–REQUIRING 1–1 (IRE1)–dependent RNA degradation (RIDD) pathway that degrades RNAs, especially for secretory proteins bound to ribosomes[3]. Notably, seeds can maintain high protein contents, suggesting a yet largely unknown modulation of protein homeostasis (proteostasis).

The cysteine proteases metacaspases (MCAs) are present in bacteria and all eukaryotes except animals[4]. In contrast to animal–specific caspases, MCAs cleave proteins after arginine (R) or lysine (K) residues, but not aspartate (D)[4]. While some organisms contain a single MCA gene (e.g., budding yeast [*Saccharomyces cerevisiae*]), land plants contain multiple MCA family members. For example, the model plant Arabidopsis (*Arabidopsis thaliana*) has nine MCAs, which are classified as type I or II (with three [MCA–I] and six [MCA–II] members; Fig. 1a) based on their structure. MCA–Is modulate pathogen–induced programmed cell death (PCD), vascular development, and protein homeostatic mechanisms, such as the clearing of protein aggregates[5–7]. The plant–specific MCA–IIs are involved in abiotic stress responses, wound–induced damage–associated molecular pattern signaling, and developmental PCD (albeit, with rather weak phenotypes)[8–11].

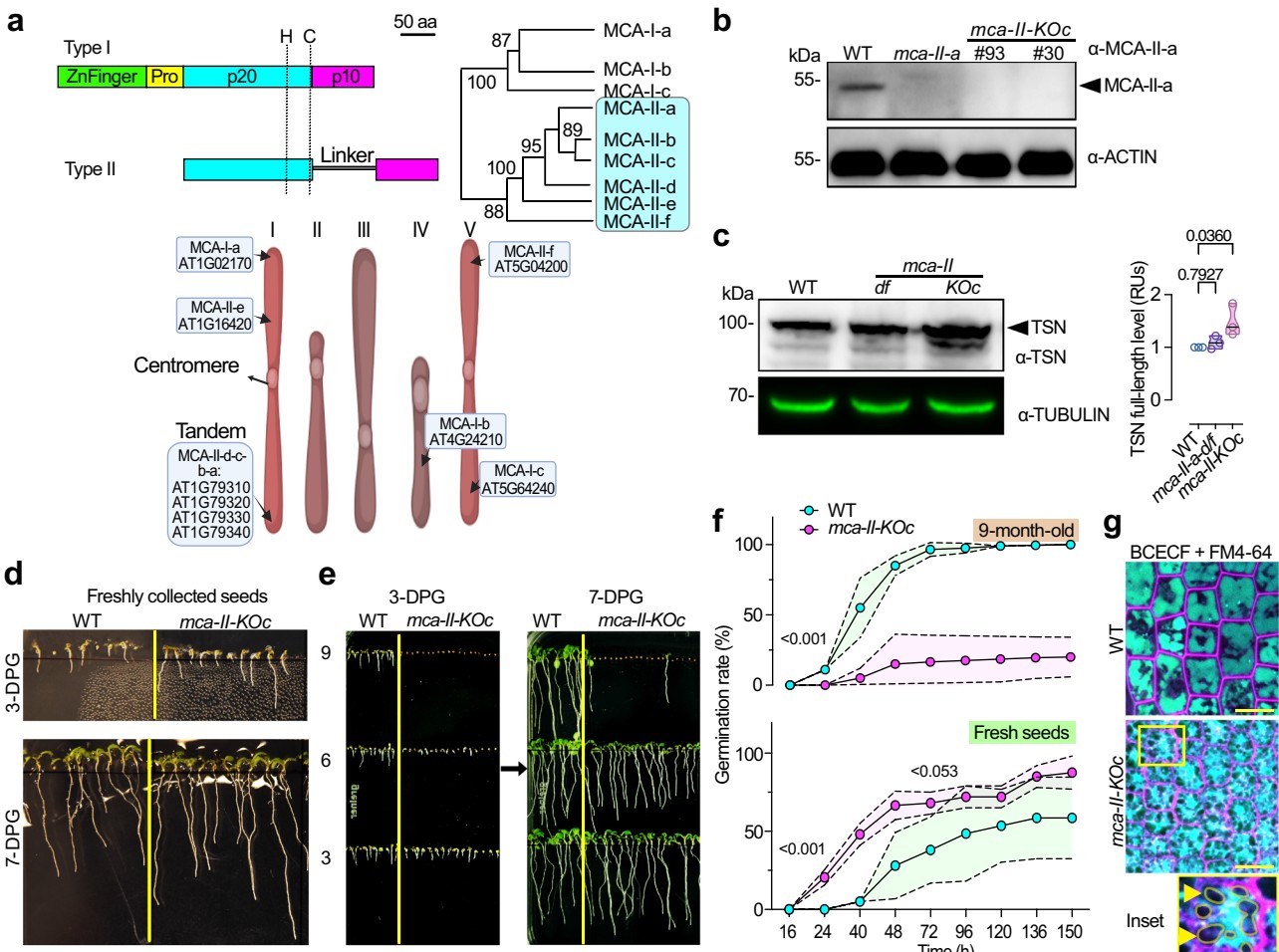

**Fig. 1 | A type II MCA depletion model affects seed physiology and vacuolar morphology. a** Schematic diagram of MCAs (top left) and phylogeny of type I and II MCAs (top right) in Arabidopsis and the nomenclature used. Bottom cartoon, chromosomal locations *MCA*s in the 5 chromosomes. Pro, prodomain; p20 and p10, large and small subunits; H, C, catalytic histidine-cysteine dyad. **b** Representative immunoblot probed with α-MCA-II-a from seedlings of the *mca-II-a* mutant and the *mca-II-KOc* (lines #93 and #30) at 7 days post germination (DPG). α-ACTIN was used as a loading control; the black arrowhead indicates MCA-II-a band (biological replicates *N* = 3, *n* = 1 technical replicate with 20 seedlings per lane). **c** Immunoblot probed with α-TSN from WT, *mca-II-df* double mutant (background used to generate the CRISPR mutants), and *mca-II-KOc* mutant seedlings. α-TUBULIN was used as a loading control; the black arrowhead indicates TSN band (*N* = 3, *n* = 1 with 20 seedlings 7 DPG per lane). Right: relative quantification of TSN band (black arrowhead, full length). *P*-values were calculated by one-way ANOVA (*N* = 3, *n* = 1 wells). **d** Representative images showing the growth of seedlings germinated from freshly collected WT and *mca-II-KOc* seeds at 3 and 7 DPG (*N* = 3, *n* = 1 with ≥ 40 seedlings in total). **e** Growth of seedlings germinated from WT and *mca-II-KOc* (3, 6, and 9 month-old) seeds at 3 and 7 DPG (*N* = 3, *n* = 1 with ≥ 40 seedlings in total). **f** Germination rate (%) from freshly collected or 9 month-old seeds of WT and *mca-II-KOc*. *P*-values were calculated by one-way ANOVA (*N* = 3, *n* = 2 replicates with a total of ≥ 278 seedlings). The first *P*-value corresponds to the 1 to 3rd point interval, while the second *P*-value corresponds to the 4th point; the magenta and light-green areas around the individual points (means) represent ± s.d. **g** Visualization of vacuoles in embryonic roots from WT and *mca-II-KOc* plants after a 2 day stratification (hydration) counter-stained with the pH-sensitive lumen dye BCECF. The styryl dye FM4-64 was used to visualize the plasma membrane (cell contours, magenta). Yellow arrowheads point to vacuole-free regions in the cytoplasm that compress the vacuolar membrane in the *mca-II-KOc* (corresponding to lipid droplets; Fig. 5b). Scale bars, 10 µm. Source data are provided as a Source Data file.

Despite their importance, the exact molecular functions of MCA-IIs and, in particular, their involvement in protein homeostatic mechanisms remain elusive. This knowledge gap presumably can be attributed to functional redundancies among MCA-IIs. In support of this assumption, four of the six MCA-II genes are localized in tandem on Arabidopsis chromosome 1, making it almost impossible to obtain double or high-order transfer (T)-DNA-based mutants (Fig. 1a).

Here, by obtaining a mutant of all six MCA-IIs, we reveal a proteostatic pathway essential for seed physiology. We show that the MCA-II-dependent proteostatic pathway operates at the ER and the lipid droplets. This pathway by impacting proteostasis at the two compartments, affects seed germination and lifespan, with potential ecological and agronomical implications.

## Results

### Type II metacaspases have redundant functions

We use the unified nomenclature for MCAs (ref. 4) as described in Fig. 1a and Supplementary Data 1, which differs from the descriptors of the TAIR database. To overcome the potential redundancies among MCA-IIs, we used clustered regularly interspaced short palindromic repeat (CRISPR)/CRISPR-associated nuclease 9 (Cas9)-mediated genome editing. First, we introduced four mutations in *mca-II-a/b/c/e* using the T-DNA double mutant *mca-II-df* as background (Supplementary Fig. 1a–c). We found that the *mca-II-d* mutant allele carried a T-DNA insertion upstream of the first start codon and showed reduced but detectable levels of *mca-II-d* mRNA (Supplementary Fig. 1c, d). To remove residual MCA-II-d, we targeted its gene through CRISPR in the *mca-II-a/b/c/e/f* mutant background. After screening ~1,000

individual plants, we obtained plants harboring single or higher-order homozygous mutations in the *MCA-II* genes in various combinations, as well as two independent lines with homozygous mutations in all six MCA-IIs. These sextuple MCA-II mutants were crossed to wild type (WT) to remove Cas9/guide RNA transgenes, and all mutations were validated in the F2 generation; we refer to these mutants hereafter as *mca-II-KOc* ("c" for *"clean line"* without transgenes; Supplementary Fig. 1e). Both *mca-II-KOc* lines are likely loss-of-function mutants, as they displayed: (1) no MCA-II-a protein as revealed with a specific antibody in immunoblots (this antibody does not cross-react with other MCA-IIs), accompanied by lower transcript levels of *MCA-II* genes, suggesting that the mutations introduced by gene editing led to nonsense-mediated decay, a process known to affect transcripts with premature stop codons[12]; (2) diminished cleavage of the MCA-II-a substrate Tudor Staphylococcal Nuclease (TSN; ref. 13–15); and (3) lower in vitro activity on an MCA-IIs fluorogenic substrate (ref. 8; Fig. 1b, c and Supplementary Fig. 1f). By using highly sensitive data-independent acquisition (DIA) proteomics (see also below), we could not detect MCA-II-a and MCA-II-f in seed proteomes of *mca-II-KOc* (Supplementary Data 2 and https://zenodo.org/records/12684164). Other MCA-IIs were not detectable in WT or *mca-II-KOc* proteomes through this approach. We should note, however, that despite not being able to detect other MCA-IIs, this by no means can be considered as lack of expression. Yet, this result further confirms that *mca-II-KOc* are loss-of-function mutants. Furthermore, MCA-I-a, which was not targeted by CRISPR, did not differ in levels between *mca-II-KOc* and WT (Supplementary Data 2 and https://zenodo.org/records/12684164; other type-I MCAs were not detected through this approach).

Remarkably, the *mca-II-KOc* plants displayed only mild developmental defects, with reduced leaf serration, slightly earlier flowering (by 4–8 days; depending on growth conditions), and earlier leaf senescence compared to the WT (Supplementary Fig. 2a). Moreover, as MCAs have been previously linked with PCD phenotypes, and considering their evolutionary link to caspases which are involved in cell death, we searched for changes in PCD. However, *mca-II-KOc* plants did not show obvious signs of compromised developmental PCD, e.g., in the root cap where MCA-IIs are expressed (Supplementary Figs. 2b and 3). Hence, MCA-IIs are likely not involved in generic cell death programs in root cap during development, at least in Arabidopsis. The involvement of MCA-IIs in more restricted, specific, and tightly controlled cell death programs (e.g., in the seed coat, endosperm, or embryo suspensor[16]) requires further investigation.

During our studies we noticed that freshly collected *mca-II-KOc* seeds germinated more quickly than WT seeds, suggesting reduced primary seed dormancy and a possible transition to the so-called "after-ripening state" in which plants attain germination competency (Fig. 1d; ref. 17). When the same batch of seeds was stored for >3 months at 4 °C (as low temperatures increase the lifespan of Arabidopsis seeds), the germination rate of *mca-II-KOc* seeds rapidly declined relative to the WT (Fig. 1e, f). Germination failure was accompanied by fragmented and irregularly shaped storage vacuoles that accumulated aggregates. This vacuolar fragmentation was associated with unknown large structures that appeared to compress and deform the vacuolar membrane (Fig. 1g). We discuss the nature of these structures below. The seed germination phenotype was mostly specific to *mca-II-KOc* and was seldom observed in the corresponding lower-order *mca-II* mutants. Even a single active MCA-II was sufficient to suppress this phenotype significantly (Supplementary Fig. 4a, b; note the expression of *MCA-IIs* and the stronger MCA-II-a association with the phenotype due to its higher expression levels in seeds and the findings described below). These results confirm our assumption that MCA-IIs show functional redundancies and further reveal the role of MCA-IIs in seeds.

## Type II metacaspases participate in a UPR-independent proteostasis pathway

As we observed storage vacuoles resembling the irregularly shaped protein-rich aggregates reported in ref. 18, we speculated that MCA-IIs modulate aspects of seed proteostasis. To identify relevant substrates of MCA-IIs, we conducted a proteomic analysis of *mca-II-KOc* and WT dry seeds (stored at 4 °C for 6 months, before *mca-II-KOc* seeds lose viability). We used a $\log_2$fold-change ($\log_2$FC) > 1 (peptide abundance average) as the criterion for enrichment between *mca-II-KOc* and WT proteins (Supplementary Data 3). Gene ontology (GO) analysis of the enriched proteins showed that *mca-II-KOc* seeds accumulate proteins residing at the ER or associated structures (i.e., lipid droplets) and proteins related to the UPR, e.g., the major ER stress-associated chaperones BiP (BINDING IMMUNOGLOBULIN PROTEIN) and disulfide isomerases (PDIs; Fig. 2a and Supplementary Data 3; refs. 19,20). We thus speculated that MCA-IIs directly cleave proteins at the ER.

To test this hypothesis, we selected MCA-II-a, -b, and -f as representative MCA-II proteins, considering that MCA-II-a and -b are very similar, and that -a and -f can be detected in seeds. We generated stable transgenic lines expressing translational fusions of these metacaspases with green fluorescent protein (GFP) under the control of their respective native promoters (i.e., sequences 2 kb upstream of the start codon). We observed a partial colocalization of GFP fluorescence signal with most of the ER sheets and associated structures (tagged with *35Spro:HDEL-CFP* for MCA-II-a and MCA-II-f) in embryonic root cells (Fig. 2b). To biochemically validate this observation, we used cell fractionation on sucrose density gradients in the presence or absence of $Mg^{2+}$, which causes a detectable increase in ER density by stabilizing its association with polysomes[21], and revealed that MCA-II-a partially co-fractionated with the ER-associated luminal protein BiP2 (Fig. 3a). In this experiment, $Mg^{2+}$ induced a shift of MCA-II-a signal by two fractions, confirming the partial association with the ER. Another protein, the BRASSINOSTEROID INSENSITIVE 1 (BRI1) which localizes at the plasma membrane, was not affected by $Mg^{2+}$ (Fig. 3a). Interestingly, MCA-II-a-GFP showed a broader signal intensity around ER sheets than CFP-HDEL or RFP-HDEL, suggesting that MCA-II-a may not be an ER luminal protein (Fig. 2b, plot profile and Supplementary Fig. 5a, b). We verified this proposition by showing that MCA-II-a was sensitive in protease-protection assays from microsomes (Supplementary Fig. 5c). We further confirmed that MCA-II-a localized on the cytoplasmic face of the ER but not in the lumen, using a split-GFP assay in the transient system of *Nicotiana benthamiana* (Supplementary Fig. 6). This assay is based on GFP auto-assembly, when two polypeptides – the "detector" GFP1-10 (residues 1-214) and the "tag" GFP11 (residues 215-230) – both non-fluorescing on their own, associate spontaneously and produce fluorescence[22]. Specifically, we observed high fluorescent signal intensity when GFP11-MCA-II-a was co-expressed with a cytoplasmic detector (i.e., GFP1-10). However, the signal was very weak when GFP11-MCA-II-a was co-expressed with a version of the "detector" residing in the ER lumen (i.e., GFP1-10-HDEL). As expected, the corresponding controls showed either high or very weak signals (i.e., GFP11 co-expressed with GFP1-10, high signal; GFP11 co-expressed with GFP1-10-HDEL, weak signal). Hence, it is unlikely that under the tested conditions, MCA-II-a and presumably other MCA-IIs enter the ER lumen. These findings show that MCA-IIs associate at least transiently with ER-related structures but do not directly access luminal proteins.

As MCA-II-a is likely excluded from the ER lumen, we aimed to test further whether ER luminal proteins are left intact. We thus performed protein degradome analysis of the seed proteome via combined fractional diagonal chromatography (COFRADIC)[23]. This approach enables the identification of proteolytic substrates, the corresponding cleavage sites, and the resulting novel N termini (*neo*-N termini) formed in vivo (Supplementary Fig. 7a-c). We confirmed lysine (K) as the preferable residue preluding the

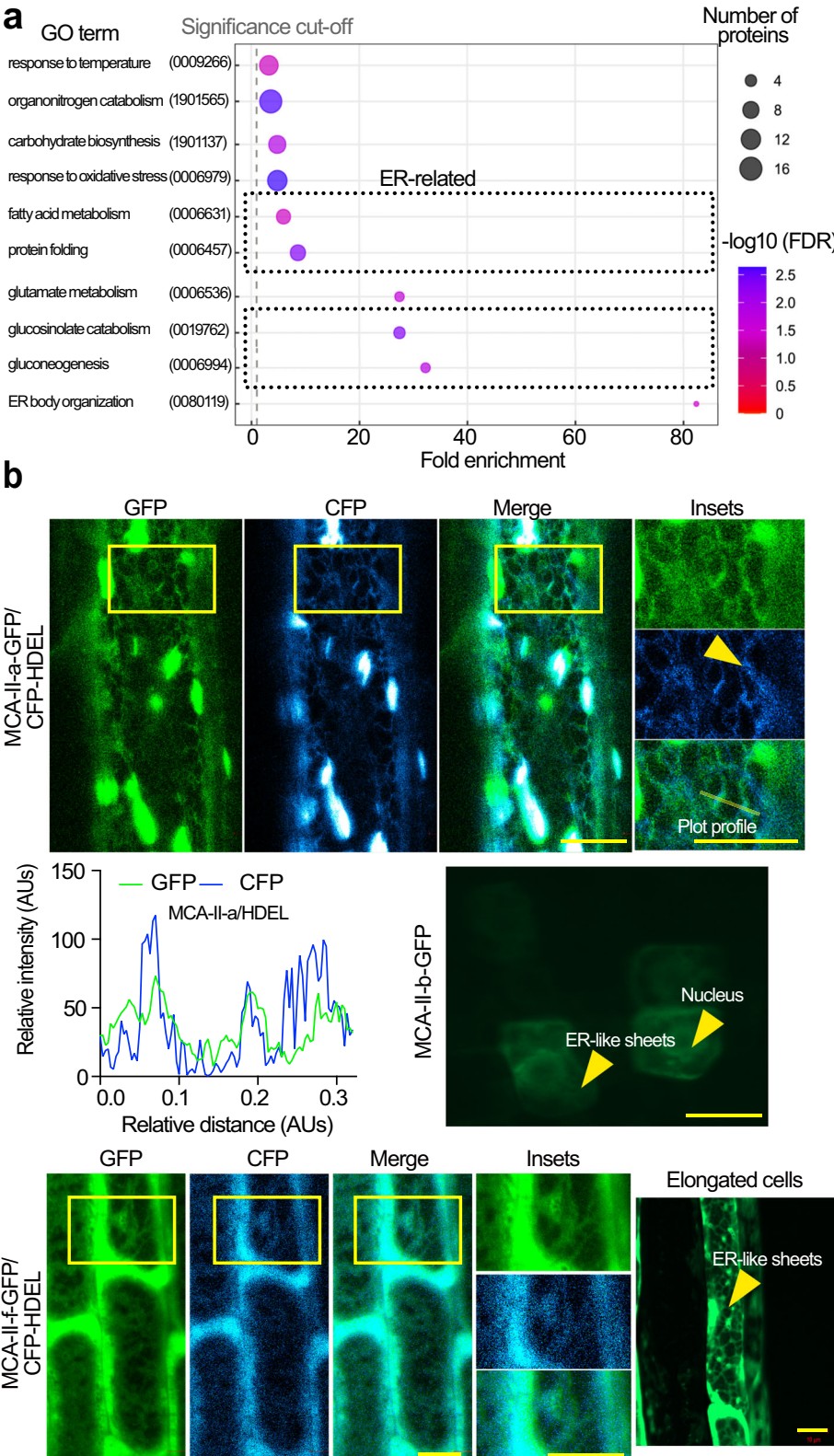

enriched *neo*-N-termini consistent with the cleavage specificity of MCA-IIs (Supplementary Fig. 7b). Furthermore, COFRADIC analysis GO terms related to seed physiology (Supplementary Fig. 7c). However, we failed to detect any of the enriched ER proteins identified in the *mca-II-KOc* seeds as direct targets of MCA-IIs by a comparative COFRADIC analysis with the WT seeds (Supplementary Data 4).

We, therefore, examined the alternative possibility that *mca-II-KOc* seeds indirectly accumulate the identified proteins in the lumen by mounting an ER stress response. This response could lead to the activation of the UPR, associated with the accumulation of BiPs and PDIs[3]. However, transcriptome deep sequencing (RNA-seq) experiments did not identify the known RNA signatures associated with the UPR in *mca-II-KOc* seeds (Supplementary Fig. 8 and Supplementary

**Fig. 2 | MCA-IIs associate with the ER. a** Gene ontology term (GO) enrichment analysis of biological processes for proteins more abundant in 6 month-old seeds of *mca-II-KO*c compared with WT (log$_2$FC ≥ 1). The black dotted box highlights relevant overrepresented and overlapping GO terms. The GO term "fatty acid metabolic process" (GO:0006631) was confirmed by the greatly diminished levels of oleosin and enlarged lipid droplets in the mutant (Fig. 5b). Furthermore, the GO terms "protein folding" (GO:0006457) and "ER body organization" (GO:00080119) were enriched and included proteins such as HEAT SHOCK 70 kDa proteins (HSP70s) and protein disulfide-isomerases (PDI5, PDI6: 3,11- and 5.0-fold, respectively), BiP 1 and 2 (3.4- and 2.6-fold, respectively). In the GO term "protein folding", the FKBP15-2 (FK506- AND RAPAMYCIN-BINDING PROTEIN 15 KD-2, immunophilin protein) was not found in WT, it is involved in ER stress sensing and accelerates protein folding. Furthermore, in the same GO term, the proteasomal subunits (GO, protein folding) such as PAG1 (20 S proteasome alpha subunit G-1: 1.3-fold) and PBA1 (20 S proteasome subunit beta 1: not found in WT) were verified using α-PAG1

and α-PBA1 antibodies in Supplementary Fig. 11g, confirming high increase of PAG1 and a smaller one for PBA1. In the term "ER body organization", PYK10 (BGLU23), BGLU18, and BGLU25 (highlighted), β-glucosidases are enriched ≥ 100-fold in *mca-II-KOc*. The GO term "gluconeogenesis" (GO:0006094) confirms the relevance of this analysis, as MCA-IIs were previously linked to this process[49]. FDR, false discovery rate. **b** Confocal images of root embryonic cells from Arabidopsis seedlings co-expressing *MCA-II-apro:MCA-II-a-GFP* (upper) or *MCA-II-fpro:MCA-II-f-GFP* (lower; elongated cells from roots also shown were due to lower levels of GFP signal the ER is better visualized) with *35Spro:HDEL-CFP* or from *MCA-II-bpro:MCA-II-b* without HDEL-CFP (middle). The yellow arrowhead in *MCA-II-apro:MCA-II-a-GFP* case denotes an ER sheet at which MCA-II-a is not localizing. Scale bars, 5 µm. The intensity plot profile of the GFP/CFP signal corresponds to intensities calculated at an ER sheet for MCa-II-a-GFP/CFP-HDEL (the region used for the plot profiling is shown in the "merged"). AUs, arbitrary units. Source data are provided as a Source Data file.

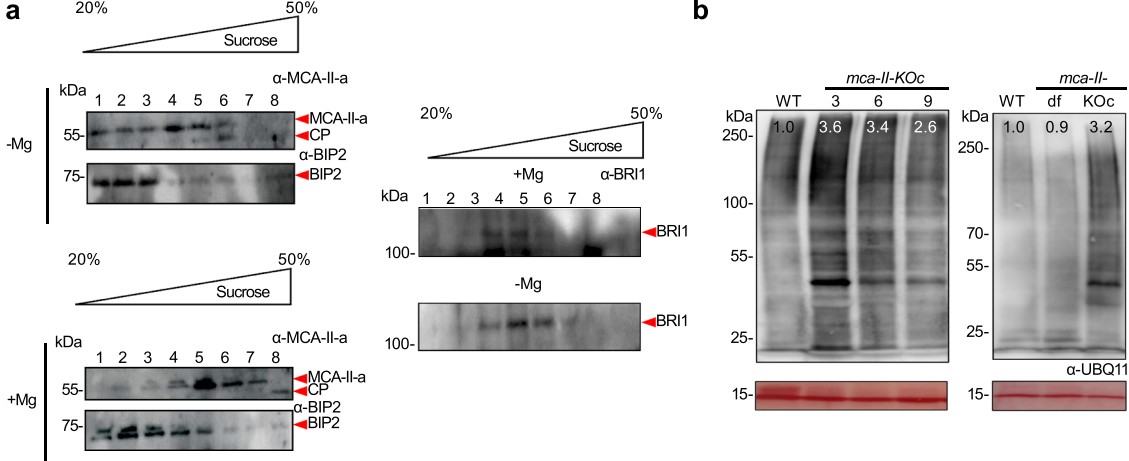

**Fig. 3 | MCA-IIs co-fractionate with the ER and decrease the accumulation of ubiquitinated proteins. a** Immunoblots probed with α-MCA-II-a, α-BiP2, and α-BRI1 from protein extracts fractionated by sucrose gradient ultracentrifugation in the presence (lower blots) or absence (upper blots) of Mg$^{2+}$. Red arrows indicate the corresponding bands. Note that MCA-II-a is known to get auto-activated by self-processing and these fragments could be detected (CP, cleavage product from autoactivated MCA-II-a; *N* = 3, *n* = 1 replicate, 7 DPG). **b** Representative immunoblot

probed with α-UBIQUITIN11 (UBQ11) from seed protein extracts (50 seeds/genotype) antibody. The numbers indicate relative levels of ubiquitinated proteins compared to Ponceau S staining at the bottom (loading control). Left blot: WT (9 month-old seeds) and *mca-II-KOc* (seeds harvested at different time points; 3, 6, and 9 month-old). Right blot: WT, *mca-II-df* (the initial background used for CRISPR), and *mca-II-KOc* with 9 month-old seeds (*N* = 2, *n* = 1 replicates).

Data 5). Furthermore, the UPR mutant *ire1a ire1b* did not recapitulate the *mca-II-KOc*-specific seed phenotype (Supplementary Fig. 9). Likewise, *mca-II-KOc* did not show changes in the levels of UPR-associated proteins at earlier time points, as well as RIDD-specific genes and proteins when compared to WT (Supplementary Data 6). Remarkably, the expression of UPR genes was reduced with seed storage time in WT (Supplementary Data 6), suggesting that other proteostatic pathways could be more relevant for long-term seed storage. Taken together, these results suggest that although *mca-II-KOc* seeds show signs of proteostatic dysregulation, they do not mount a typical UPR.

Based on these results, we hypothesized that MCAs may function in alternative pathways of proteostasis that could somehow converge in the ER. We thus tested whether MCA-IIs interact with proteins involved in proteostasis using the two very similar MCA-II-a and -b and their inactive **P**roteolytically-**D**ead variants (MCA-II-a/b$^{PD}$, with their catalytic Cys replaced with Ala) as baits for affinity purification followed by liquid chromatography-tandem mass spectrometry. These assays showed that MCA-II-a/b co-precipitated with two chaperones involved in proteasome assembly (Supplementary Fig. 10a, b and Supplementary Data 7; 2 out of 32 proteins in the proteasome assembly; REGULATORY PARTICLE TRIPLE-A ATPASE 5A-RPT5a and

AT5G45620-SS regulatory sub 13a). Mutating the catalytic Cys of these two MCAs (MCA-II-a/b$^{PD}$) facilitated these interactions, likely because these mutants remained bound to their substrates for longer as it has been suggested for other (pseudo-)proteases (Supplementary Data 7; ref. 24). In accordance, the mutant MCA-II-a$^{PD}$ bait, but not its WT version, was associated with (poly-)ubiquitin (Ub)-conjugated proteins in *N. benthamiana* transient expression system (Supplementary Fig. 10c, d). Ub-proteins are usually recognized by the proteasome for degradation. Accordingly, Ub-conjugated proteins were more abundant in *mca-II-KOc* relative to WT (Fig. 3b; ~3-fold). Moreover, subunits of the proteasome accumulated to higher levels in *mca-II-KOc* compared to WT (Supplementary Data 3). We also observed that some regulatory subunits of the 26 S proteasome accumulated in *mca-II-KOc*, the PAG1 (20 S proteasome alpha subunit G- 1: 1.3-fold), and PBA1 (20 S proteasome subunit beta 1, increased only after germination Supplementary Fig. 11a-c). This biochemical phenotype is reminiscent of proteasomal receptor mutants, e.g., mutants of the REGULATORY PARTICLE NON-ATPASE 10 (*rpn10*; ref. 25). Furthermore, *mca-II-KOc* was insensitive to MG132, a peptide aldehyde that reversibly inhibits the proteasome, as was reported for several proteasomal mutants (Supplementary Fig. 12; ref. 26). Notably, active MCA-II-a did not cleave K48 tetra-Ub isopeptide linkages, which

represent the most abundant and canonical degradation signals (Supplementary Fig. 13; ref. 27). This finding argues against the possibility that MCA–IIs directly remove Ub–moieties from proteins (i.e., not acting as the cysteine proteases known as "deubiquitinases"). Taken together, these results suggest that MCA–IIs participate in a proteostatic pathway that despite being independent of UPR, still involves the proteasome.

## Type II metacaspases interact with CDC48 and the proteostatic lipid droplet-associated degradation pathway

As the UPR was not induced while proteasome–dependent protein degradation still was compromised in *mca–II–KOc* seeds, we hypothesized that Arabidopsis seeds rely on an alternative pathway for protein homeostasis. We focused on ER–associated degradation (ERAD), as this process would still promote the retrotranslocation of Ub–proteins from the ER to the cytosol for degradation by the proteasome, much like the UPR[28,29]. We assumed that ERAD would fit well in the context of seed proteostasis, potentially by allowing the excess accumulation of secreted seed storage proteins (e.g., 2 S albumins, late embryogenesis abundant proteins, and 12 S globulins). These proteins nurture and protect the embryo. On the contrary, the UPR pathway could compromise the production of these secreted proteins by eliciting their translational suppression through RIDD[28,29]. Furthermore, ERAD by directly targeting UPR components may downregulate this pathway[30,31], for example, in older seeds (this hypothesis is supported by Supplementary Data 6).

AAA ATPase CELL DIVISION CYCLE 48 (CDC48; valosin–containing protein [VCP] in vertebrates) is a major ERAD component that exports proteins from the ER for proteasomal delivery and regulates proteasome activity, functioning as an "unfoldase/segregase"[29]. In budding yeast, the sole type I MCA (Mca1p) interacts with CDC48 to control the accumulation of protein aggregates[32]. We verified that CDC48 colocalized with MCA–II–a in embryonic roots harboring the transgenes *RPS5apro:MCA–II–a–mNeon* and *35Spro:mCherry–CDC48a*; the two proteins interacted also as shown using a quantitative in vivo proximity ligation assay (PLA) in the root cells of stable lines expressing *RPS5apro:His–Flag(HF)–MCA–II–tagRFP* (Fig. 4a–c). We corroborated the partial colocalization of MCA–II–a with CDC48 using specific antibodies (Fig. 4d, Pearson *r* = 0.44). Interestingly, under the same setting, MCA–II–a colocalized with CDC48 on structures produced from the ER or that are associated with it, *viz.* lipid droplets. From these assays, we conclude that in seeds, MCA–II–a, and CDC48 occasionally colocalized to lipid droplets, suggesting their transient association (Fig. 4e, arrowheads).

Given that the lipid droplet–associated oleosins, required for the structural integrity of lipid droplets, increase seed viability[33], and as we observed, seed-related phenotypes for *mca–II–KOc*, we focused on the specialized type of ERAD pathway named lipid droplet–associated degradation (LDAD). This pathway removes oleosins from lipid droplets in seeds, facilitating their breakdown and thus the release of fatty acids[33]. CDC48 colocalized and interacted at least transiently with MCA–II–a, and its homologs (e.g., CDC48E) were 4.1 times more abundant in the *mca–II–KOc* proteome dataset compared to WT, implying a compensatory function (Supplementary Data 3). We thus asked whether MCA–IIs modulate CDC48, and to address this, we used LDAD activity as a readout. Lipid droplets fuse to form larger droplets in the absence of oleosins, as has been observed in loss-of-function oleosin mutants[34]. We observed a reduced abundance of oleosin levels in *mca–II–KOc* in western blots, proteomics, and protein gels by >30-fold (Fig. 5a, Supplementary Fig. 14 and Supplementary Data 8; mainly OLE1). Accordingly, *mca–II–KOc* showed an increased size of lipid droplets compared to WT (Fig. 5b). This phenotype was partially complemented through the expression of *RPS5apro:MCA–II–a–mNeon* in *mca–II–KOc* (Fig. 5a, b; "Com"). Notably, *mca–II–KOc* did not show significantly different fatty acid levels and only slight differences in

composition (Supplementary Fig. 15 and Supplementary Data 9; mainly lower content in 18:2 and higher content in 18:3), suggesting that LDAD does not substantially contribute to the metabolism of lipid pools in seeds which could affect germination; we cannot discount, however, an effect of local lipid distribution or profiles on membranes in germination. Hence, we suggest that the structures compressing the vacuole in seeds are likely the lipid droplets. Furthermore, as lipid droplets can physically associate with the ER, we cannot exclude the possibility of additional membranes (e.g., ER sheets) also contributing to this deformation. Altogether, these results suggest that MCA–IIs may modulate the dynamics of lipid droplets, which store fatty acids and thus regulate energy reserves[33].

Next, we aimed to address how MCA–IIs could modulate directly the LDAD pathway. During LDAD, the CDC48 adapter PUX10 (ubiquitination regulatory X [UBX] domain–containing 10), one of the 16 PUX adapters in Arabidopsis defined by a Ub–like UBX domain, specifically recognizes Ub–oleosin for degradation. PUX10 also harbors a ubiquitin–associated (UBA) domain at its N terminus[35]. In the absence of PUX10 (i.e., in loss-of-function *pux10* mutants), oleosin accumulates (ref. 33 and results below). We observed that in the DIA proteomics, PUX10 showed increased levels in *mca–II–KOc* compared to the WT (Supplementary Fig. 16a, b and Supplementary Data 10); we did not find a similar regulation of other PUX proteins that we could detect in seeds (Supplementary Fig. 16a–c and Supplementary Data 10; 10 out of 16 PUXs were detected). We also observed that the levels of PUX10–GFP expressed under the native PUX10 promoter in immunoblots probed with a-GFP were significantly higher in *mca–II–KOc* than in WT (Fig. 6a). Furthermore, PUX10 was significantly more stable in *mca–II–KOc* as revealed by following its degradation rate through immunoblots and live-cell imaging of root cells in the presence of the translation inhibitor cycloheximide and the proteasome inhibitor MG132 (Fig. 6a–c). By introducing *PUX10pro:PUX10–myc* in WT, or the *mca–II–KOc* expressing *MCA–II–a–GFP* or the corresponding inactive variant *MCA–II–a^{PD}–GFP*, we confirmed that PUX10 stability depends on active MCA–II–a (Fig. 6d).

This result could imply direct interactions between PUX10 and MCA–II–a. We confirmed that MCA–II–a and PUX10 colocalize and interact by Förster resonance energy transfer–sensitized emission (FRET–SE) and PLA assays, in embryonic roots with the transgenes *RPS5apro:MCA–II–a–tagRFP* and *PUX10pro:PUX10–GFP* (Fig. 7a–d). Noteworthy, MG132 application increased FRET efficiency, suggesting that MCA-II-a remains associated with PUX10 for longer when the proteasome is inactive (Fig. 7d). We could also detect FRET signal in the nucleus in the presence of MG132 (Fig. 7d, "Nuc") where PUX10-GFP and MCA-II-a-tagRFP also localized. This signal suggests the transient association of PUX10/MCA-II-a in the nucleus or at a perinuclear region. Accordingly, we observed an anti-correlation between signal intensities of fluorescently-tagged MCA-II-a and PUX10 in roots, suggesting that MCA-II-a targets PUX10 and reduces its levels (Fig. 7d and Supplementary Fig. 17). We further confirmed the specific cleavage of PUX10 by MCA–II–a but not by MCA–II–a^{PD} variant in *N. benthamiana* (Supplementary Fig. 18a). This cleavage led to the accumulation of an N-terminal fragment of PUX10 that was associated mainly with MCA–II–a and very weakly with MCA–II–b or MCA–II–f in a co-immunoprecipitation assay in *N. benthamiana* (Supplementary Fig. 18a–c; note the lack of CDC48 cleavage under the same conditions). Altogether, our results suggest that MCA–IIs regulate LDAD by interacting and cleaving PUX10.

## Metacaspases and PUX10 regulate ERAD to confer seed longevity

The lack of PUX10 cleavage in *mca–II–KOc* would be expected to lead to increased retention of CDC48 at the lipid droplets. This hypothesis was validated by the observation that, in contrast to WT, in *mca–II–KOc*, mCherry–CDC48 decorated large intracellular structures

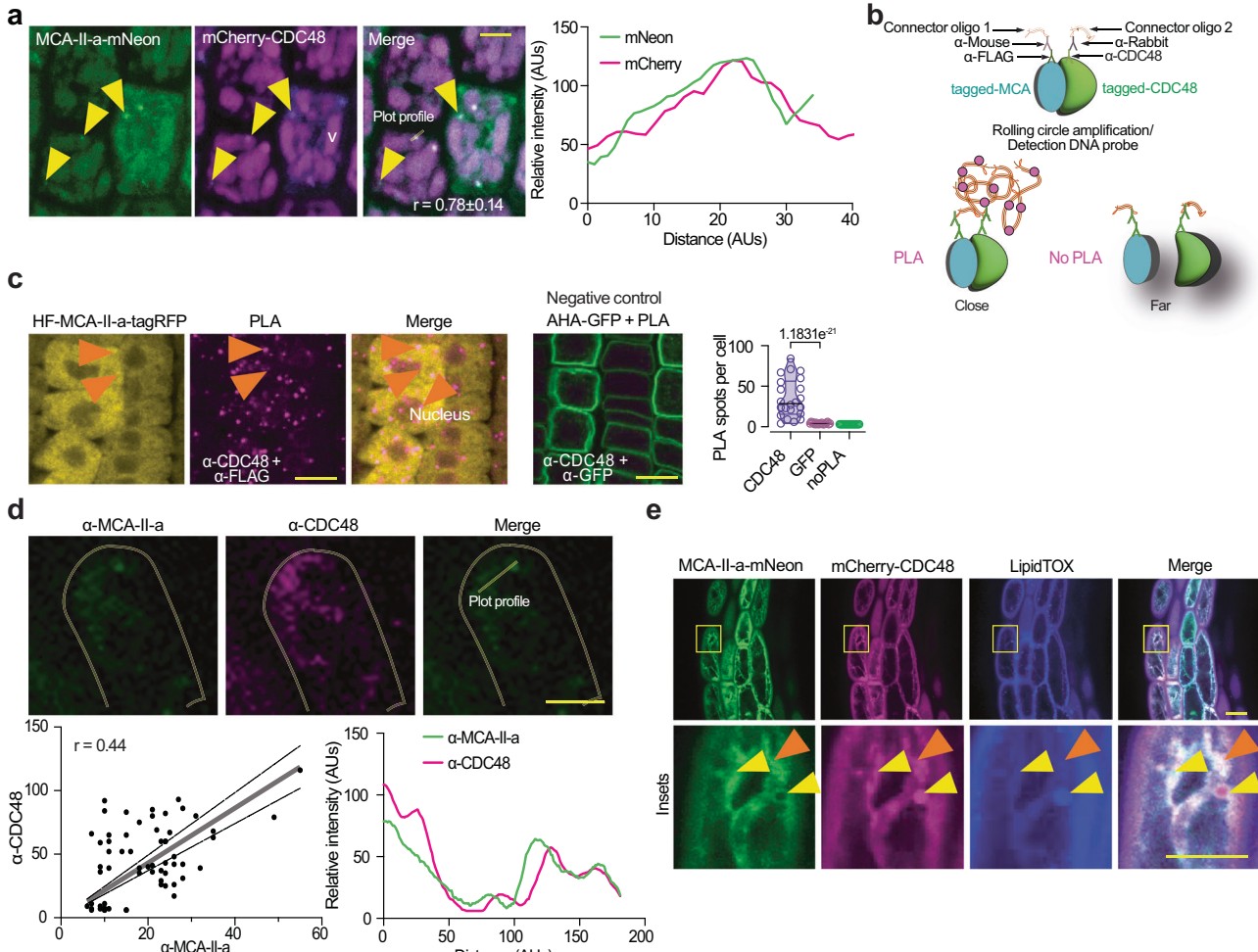

**Fig. 4 | MCA-IIs associate with CDC48. a** Confocal micrographs of embryonic roots after a 2 day stratification, coexpressing *RPS5apro:MCA-II-a-mNeon* and *35Spro:mCherry-CDC48*. Pearson's correlation coefficients (r) estimate colocalization between MCA-II-a and CDC48 and are shown on the merged micrograph. The yellow arrowheads denote sites of colocalization between CDC48 and MCA-II-a in puncta. The experiment was repeated more than five times (v, vacuole). Scale bars, 5 μm. Lower: pixel intensity plot profile of mScarlet/mNeon signals on puncta (AUs, arbitrary units). The region used for profiling is shown in the inset. **b** Proximity Ligation Assay (PLA) approach principle (for < 40 nm protein distance, details in the methods). **c** Confocal micrographs of PLA signal (PLA foci denoted with orange arrowheads; signal in the nucleus was also detected and is denoted in the "merge") indicating the interaction between HF-MCA-II-a-tagRFP (HF: 6xHis-3xFLAG tag) with CDC48 (α-CDC48) from epidermal cells of meristematic cells from root (5 DPG). PLA of the P-ATPase AHA1-GFP with CDC48 was used as a negative control (α-GFP and α-CDC48). Scale bars, 10 μm (*N* = 3, *n* = 10 seedlings, multiple cells).

Right: quantification of PLA-positive foci in cells. *P-values* were calculated by one-way ANOVA (*N* = 3, *n* = 8 cells/sample). **d** Confocal micrographs from embryonic root epidermal cells probed with α-MCA-II-a and α-CDC48 (*N* = 4, *n* ≥ 3 roots per replicate 5 DPG). Cell contours are shown (dim yellow). Scale bar, 5 μm. The charts below show intensity correlations of fluorescent signals using the Pearson correlation coefficient (r), and the intensity plot profile of the signal detected by α-MCA-II-a/α-CDC48 (right; the region used for the plot profiling is shown in the "merged"). The dashed lines show 95% confidence intervals. **e** Confocal micrographs of embryo hypocotyl cells after a 2 day stratification showing CDC48 colocalization with MCA-II-a from lines co-expressing *RPS5apro:MCA-II-a-mNeon* and *35spro:mCherry-CDC48a* (radicles produced similar results) counterstained with LipidTOX (lipid droplet staining; cyan). The yellow arrowheads denote colocalization between fluorescent signals, while the orange arrowhead denotes lack of colocalization. Scale bars, 5 μm (*N* = 3, *n* = 1 seedlings per replicate). Source data are provided as a Source Data file.

reminiscent of lipid droplets, whereas its ER-like signal decreased (Fig. 8a, left). To corroborate this result, we performed immunostainings in root cells with a-CDC48/a-BIP2 (Fig. 8a, right). While most of the CDC48 signal localized in ER sheets in WT or the *pux10* mutant, this was not evident in *mca-II-KOc*. Using ultracentrifuge fractionation analyses, we confirmed that CDC48 in *mca-II-KOc* was not associated with the ER as extensively as in WT or the *pux10* mutant (Fig. 8b; note the lack of CDC48 signal in *mca-II-KOc* fractions 2-3 where BIP2 accumulated mostly). As expected, in *pux10* mutants CDC48 did not show an association with lipid droplets (Fig. 8a, b; usually appearing in fraction 1). Interestingly, immunostaining also revealed that in *mca-II-KOc* CDC48 accumulated in the perinuclear region and in some cells at the plasma membrane (Fig. 8a). This finding implies that

*mca-II-KOc* accumulated significant amounts of damaged proteins at these two cellular regions. Hence, excessive LDAD with the simultaneous absence of MCA-IIs could lead to the retention of CDC48 at the lipid droplets reducing ERAD capacity.

To corroborate these results, we first compared the sensitivity to CB-5083, a specific inhibitor of CDC48[35], of WT and *mca-II-KOc*. Indeed, the loss of function of MCA-II enhanced the sensitivity of young seedlings to CB-5083, as the swollen root tips and shorter roots of *mca-II-KOc* indicate (Fig. 9a). Furthermore, at parts of the ER in the *N. benthamiana* transient expression system, MCA-II-a interacted with the ERAD cytoplasmic component DEGRADATION OF ALPHA2 10 (DOA10)-like E3 ligase (Supplementary Fig. 19; ref. 36). These results suggested a possible association of MCA-II-a with cytoplasmic

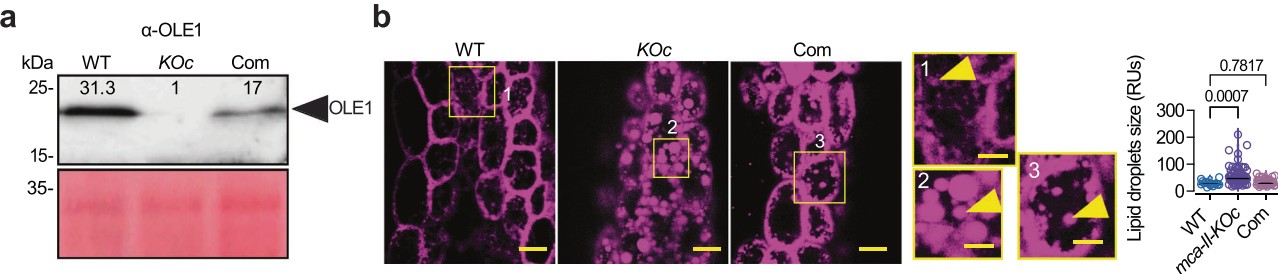

**Fig. 5 | MCA-IIs regulate the dynamics of lipid droplets.** Representative immunoblots probed with an α-OLE1 from seed proteins of WT, *mca-II-KOc* (KOc), and the Com line (*MCA-II-apro:GFP-MCA-II-a*; 3 month-old seeds). The numbers indicate relative levels compared to Ponceau S staining at the bottom (loading control; *N* = 3, *n* = 1 with 50 seeds/lane). **b** Representative confocal micrographs from WT, *mca-II-KOc*, and Com line counterstained with LipidTOX in the embryonic hypocotyl regions after a 2 day stratification. Insets (denoted 1 to 3) show details of lipid droplets; examples of lipid droplets are denoted with yellow arrowheads. Scale bars, 5 μm (1 μm for insets). Right, measurement of lipid droplet size. *P*-values were calculated by ordinary one-way ANOVA (*N* = 3, *n* ≥ 15 embryonic hypocotyls). RU, relative units. Source data are provided as a Source Data file.

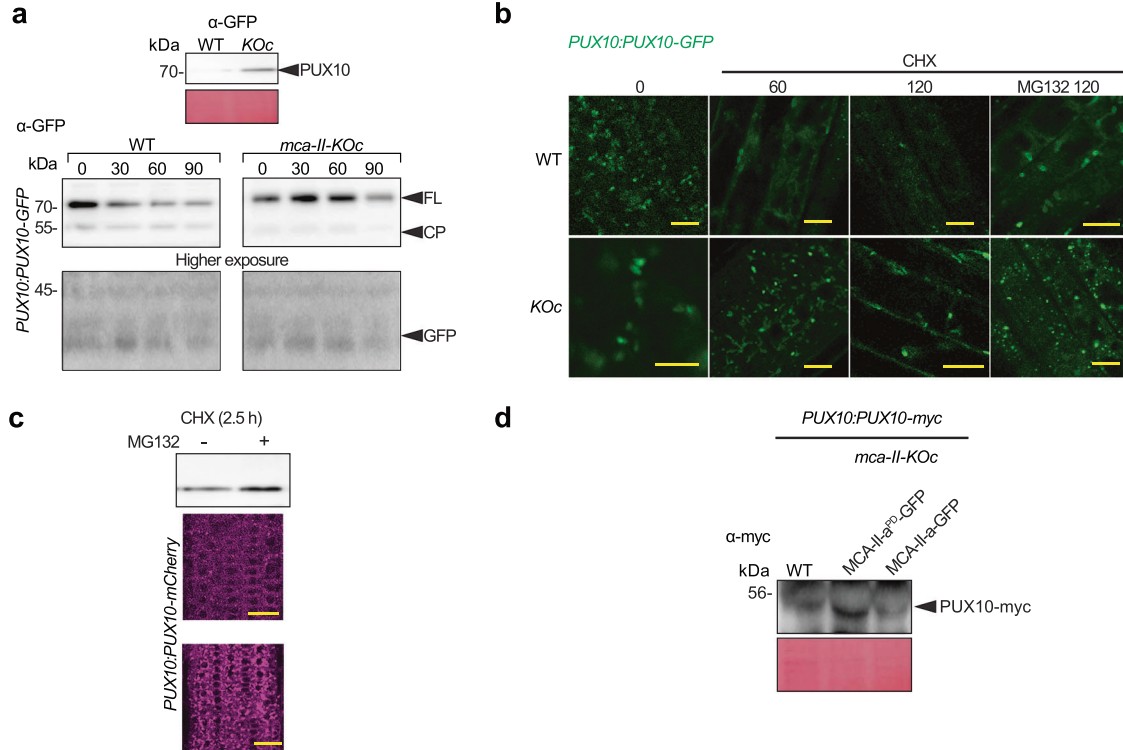

**Fig. 6 | MCA−IIs regulate the levels of PUX10 in cells. a** Upper: immunoblot of protein extracts probed with α-GFP from seedlings expressing *PUX10pro:PUX10-GFP* in WT and *mca-II-KOc* background. Lower: PUX10-GFP decay in WT or *mca-II-KOc* following CHX treatment (time points after CHX addition are denoted in min; *N* = 3, *n* = 2 replicates with ≥10 seedlings per lane 5 DPG). FL, full-length PUX10; CP, cleavage product (minus N-terminus of PUX10). **b** Confocal images from root epidermal cells of *PUX10pro:PUX10-GFP* seedlings treated with 50 μM of CHX and MG132 for various times (2 DPG; time points after CHX addition are denoted in min). Scale bars, 50 μm (*N* = 3, *n* = 1 replicate with 3 seedlings each). **c** Immunoblot probed with α-GFP from seedlings expressing *PUX10pro:PUX10-GFP* in WT in the presence or absence of MG132, and representative confocal images of embryonic roots from seedlings expressing *PUX10pro:PUX10-mCherry* in WT (upper) and *mca-II-KOc* (lower). Scale bars, 7 μm. **d** Immunoblot of protein extracts with α-myc from embryonic roots expressing *PUX10pro:PUX10-myc* in WT and *mca-II-KOc* background complemented with MCA-II-a-GFP or the corresponding inactive variant (PD; *N* = 1, *n* = 1 replicate with ≥10 seedlings per lane 5 DPG).

pathways of ERAD where CDC48 acts. We observed that *mca−II−KOc* and to a lesser extent *pux10*, showed enhanced responses to drugs or treatments that plants with compromised ERAD are hypersensitive to (Supplementary Fig. 20a; e.g., kifunensin, and thapsigargin; ref. [37]). More specifically, in kifunensin treatment, we observed swollen and shorter root tips in *mca-II-KOc* seedlings. Similarly, *mca-II-KOc* roots were smaller when treated with thapsigargin. The introduction of *pux10* mutation into *mca−II−KOc* reverted this sensitivity. Interestingly, the *pux10 mca−II−KOc* mutant showed increased sensitivity to the more general inhibitor of proteostasis tunicamycin (Supplementary Fig. 20b). These results suggest that PUX10 and MCA-IIs function

in ERAD and additional proteostatic pathways (see below). Furthermore, when proteostasis of seedlings is challenged by pharmacological means, *mca-II-KOc* fail to respond properly suggesting that their overall proteostasis is compromised.

To further corroborate the link of MCA−IIs to ERAD, we expressed BRI1 or the mutant allele Bri1−9 (S662F substitution within the BR binding domain of BRI1) under the RPS5a promoter to promote embryonic root expression in WT and *mca−II−KOc* (*RPS5apro:Bri1−9−GFP*). BRI1 is a leucine-rich-repeat (LRR) receptor-like kinase (RLK) that functions as a cell surface receptor for brassinosteroids (BRs), and the corresponding *bri1-9* mutant allele is semi-

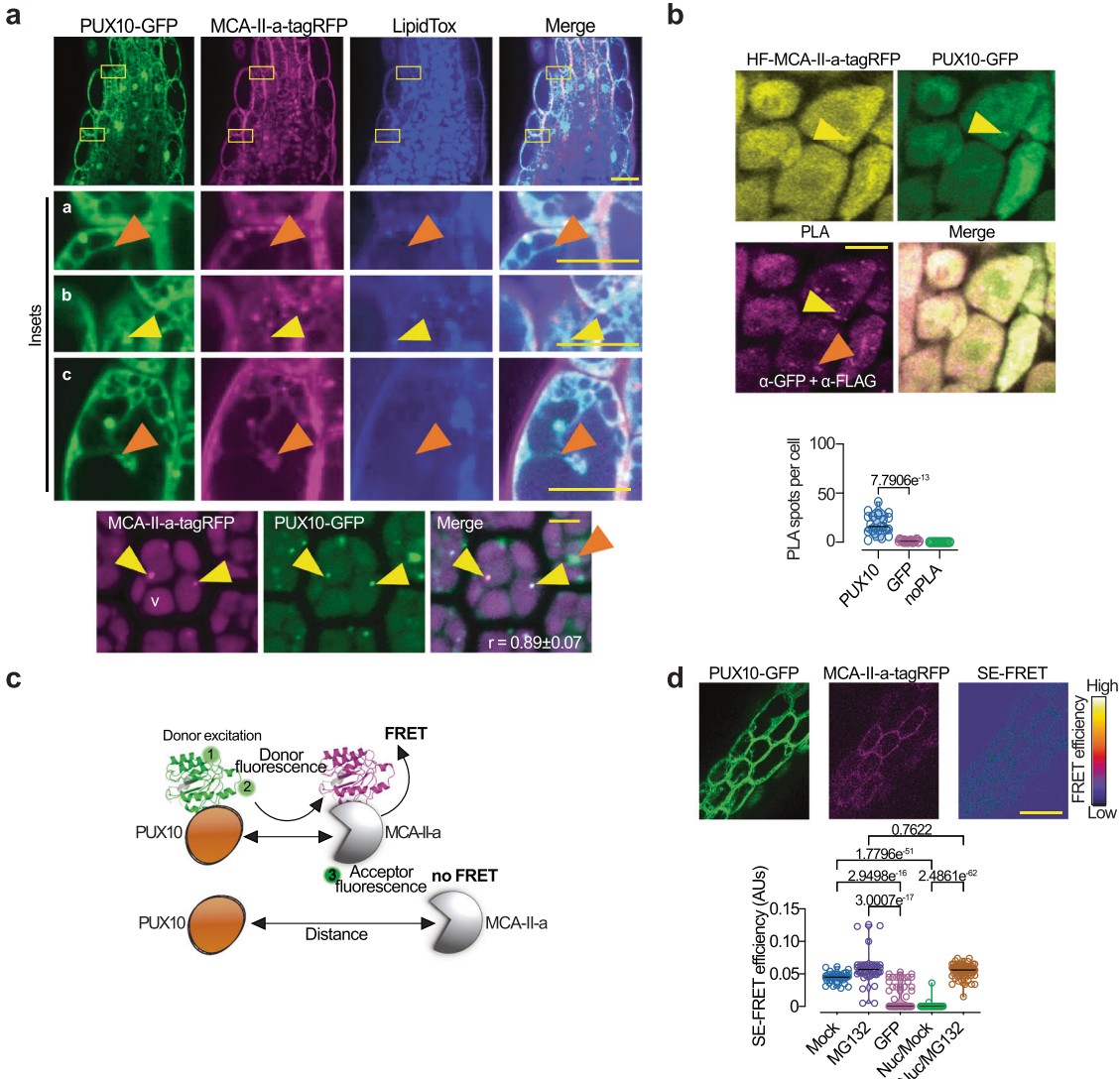

**Fig. 7 | MCA–IIs directly interact with PUX10 in cells. a** Confocal micrographs of lipid droplets stained with LipidTOX in the hypocotyl region of embryos after 2 days stratification of a line co-expressing *RPS5apro:MCA-II-a-tagRFP* and *PUX10-pro:PUX10-GFP*. In the insets denoted with "a", orange arrowheads show a lack of colocalization between MCA-II-a/PUX10 with LipidTOX; "(**b**)", yellow arrowheads show the MCA-II-a/PUX10 colocalization with LipidTOX; "(**c**)", same as in "(**a**)", showing that lack of colocalization with relatively larger droplets. Scale bars, 10 μm. The micrographs on the bottom show a surface *Z*-axis of cells, for better visualization of the PUX10 droplets. The yellow arrowheads show colocalized PUX10/MCA-II-a, while the orange arrowhead shows a lack of colocalization (note the increased signal intensity of PUX10 in this instance). The Pearson correlation coefficient (r) represents the colocalization between GFP and tagRFP signals in droplets. Scale bars, 10 μm (*N* > 5, *n* = 1 replicate with ≥10 seedlings). **b** Representative confocal micrographs (lower) of PLA signal (PLA foci denoted with magenta) indicating the interaction between HF-MCA-II-a-tagRFP (HF: 6xHis-3xFLAG tag) with PUX10-GFP (α-FLAG and α-GFP) from epidermal cells of meristematic cells from root. Yellow arrowheads denote detectable PUX10/MCA-II-a signals, while the orange arrowhead denotes submicroscopic PUX10/MCA-II-a signals that produce PLA (i.e., interactions outside droplets). Scale bar (in "PLA" for visibility), 10 μm. Lower chart: corresponding quantification of PLA-positive foci. *P-values* were calculated by one-way ANOVA (*N* = 3, *n* = 1 replicate with multiple cells). **c** The "sensitized emission" FRET (FRET-SE) approach, where the emission spectrum of the donor (1) overlaps with the excitation spectrum of the acceptor (2), and if the distance between the two molecules is sufficiently short (i.e., connoting association), energy is transferred (3). **d** Confocal micrograph showing FRET-SE intensity between PUX10-GFP and MCA-II-a-tagRFP from root epidermal cells. Scale bar, 5 μm. Bottom graph: FRET-SE between PUX10-GFP and MCA-II-a-tagRFP in the absence (mock) or presence of 50 μM MG132. GFP represents the negative control (free GFP; lines co-expressing GFP with MCA-II-a-tagRFP). *P-values* were calculated by ordinary one-way ANOVA (*N* = 2, *n* = 2 roots ≥ 7 cells 5 DPG). Source data are provided as a Source Data file.

dwarfed[37]. Compared to BRI1, the mutant Bri1–9 is relatively unstable and gets trapped by ERAD at the ER to undergo proteasomal degradation. When ERAD malfunctions, Bri1–9 escapes from the ER lumen to the plasma membrane[37]. In embryonic *mca–II–KOc* roots, Bri1–9–GFP was overall more abundant and could localize more efficiently at the plasma membrane (Fig. 9b and Supplementary Fig. 21 for rosette leaves). Furthermore, the septuple mutant *mca–II–KOc bri1–9* showed higher stability of Bri1-9 in immunoblots compared to the *bri1–9* single

mutant, and deficiency for MCA-IIs could partially rescue the dwarf phenotype of *bri1–9* (Fig. 9c, d).

We further asked whether the retention of CDC48 in lipid droplets could affect seed physiology. We reasoned that the depletion of CDC48 from the ER in *mca–II–KOc* would aggravate a chronic type of ER stress, that could eventually lead to cell death[35]; this is in line with the aforementioned accumulation of ubiquitinated proteins (Fig. 3b). We thus measured seed viability using

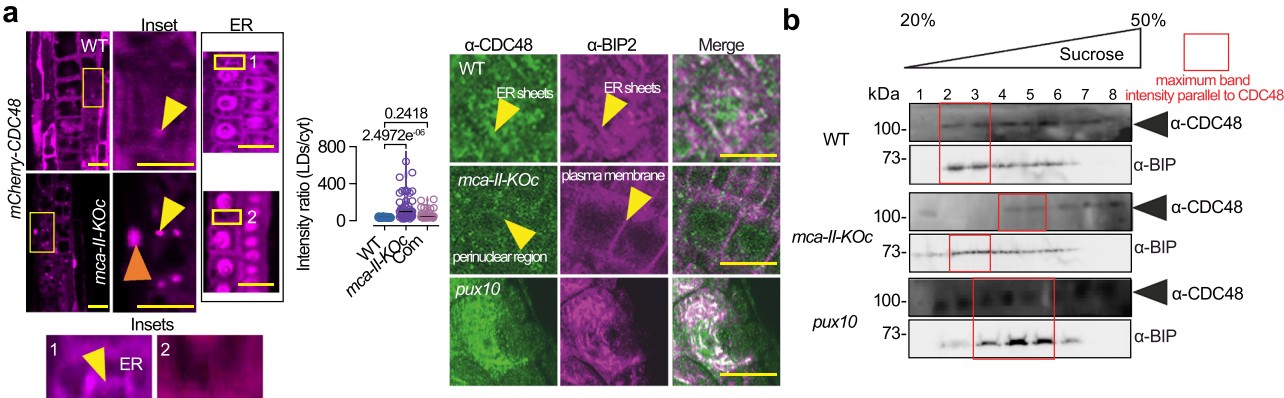

**Fig. 8 | MCA–IIs regulate the localization of CDC48 in cells. a** Confocal micrographs of embryonic root cells after 2 days stratification, from WT and *mca-II-KOc* harboring *35Spro:mCherry-CDC48a*. Scale bars, 10 μm. The yellow arrowheads denote structures reminiscent of lipid droplets, while the red arrowheads denote the nucleus. The two insets show details of CDC48 localization in WT and *mca-II-KOc* ("1" and "2", respectively; *N* = 2). Middle (chart): quantification of the intensity ratio between the mCherry-CDC48 signal at the lipid droplets (LDs) and cytoplasm in WT, *mca-II-KOc*, and "Com" (complementation, *MCA-II-apro:MCa-II-a-GFP mca-II-KOc*). *P*-values were calculated by one-way ANOVA (*N* = 3, *n* = 8 cells/sample). Right: confocal micrographs of embryonic root cells probed with α-CDC48 and α-BIP2

(for ER signal), from WT, *mca-II-KOc*, and *pux10*. Note the increased puncta signal in *mca-II-KOc*; the plasma membrane and perinuclear signals observed in this mutant are also denoted (yellow arrowheads). Scale bars, 10 μm (*N* = 2, *n* = 2 replicates with 3 roots in each, after 2 days stratification). **b** Immunoblots probed with α-CDC48/α-BIP2 from WT, *mca-II-KOc*, and *pux10* protein extracts from seedlings fractionated by sucrose gradient ultracentrifugation in the absence of Mg²⁺. The red rectangular denotes maximum band intensity; note the slight band signal offset for *mca-II-KOc* compared to α-BIP2 signal (*N* = 2, *n* = 1 replicate 5 DPG). Source data are provided as Source Data file.

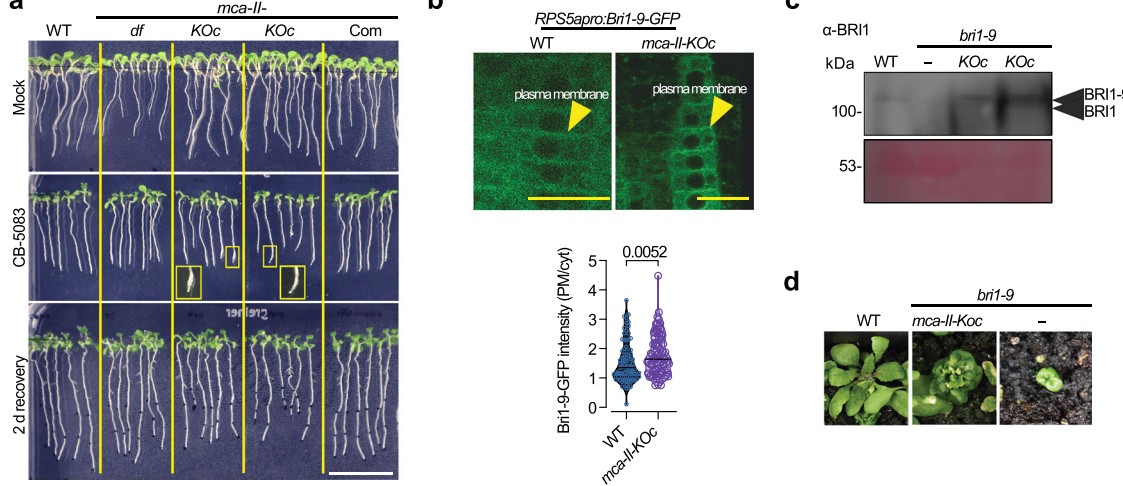

**Fig. 9 | MCA–IIs regulate the ERAD pathway. a** Images of the indicated genotypes, which were mock-treated (DMSO), treated with a CDC48 inhibitor (CB-5083, 2 μM), and following a 2 day recovery (9 day-old seedlings) from CB-5083 treatment on DMSO-containing plates. Note the swelling root tip phenotype and the shorter root (after 2 d recovery) observed in *mca-II-KOc* (two lines), which is indicative of hypersensitivity to CB-5083 (insets). Scale bar, 20 mm (*N* = 3, *n* = 8–10 seedlings/ genotype 7 DPG). **b** Representative confocal micrographs from WT or *mca-II-KOc RPS5apro:Bri1-9-GFP* embryonic roots. The plot at the bottom shows the quantification of the ratio of Bri1-9-GFP signal intensity on the plasma membrane compared

to the cytoplasmic signal ("cyt"). *P*-value, two-tailed Mann Whitney test (*N* = 3, *n* = 8-10 seedlings/genotype 2 days after stratification). **c** Representative immunoblot probed with α-BRI1 from WT, *bri1-9* and the septuple *mca-II-KOc bri1-9* mutant (*N* = 2, *n* = 1 replicate with 1 seedling root per lane 2 days after stratification). **d** Representative phenotypes of WT, *bri1-9*, and the septuple *mca-II-KOc bri1-9* mutant from a double-blind experiment where the genotype-phenotype link was established independently. Scale bar, 10 mm. Source data are provided as a Source Data file.

tetrazolium staining and found increased cell death in normally aged or exposed to the short-term controlled deterioration treatment (CDT) *mca–II–KOc* seeds (Fig. 10a). We could partially complement this phenotype using the constructs *RPS5a:MCA–II–a–GFP*, *35Spro:MCA–II–a–GFP* or *MCA–II–a:MCA–II–a–GFP*, but not the corresponding inactive variant (MCA–II–aᴾᴰ–GFP; Supplementary Fig. 22a–d). On the contrary, the lack of PUX10 did not lead to a similarly compromised seed–longevity or sensitivity to CDT (Supplementary Fig. 22e). Most importantly, the septuple *mca–II–KOc pux10* showed a similar to WT CDT phenotype, confirming also

genetically the interaction of MCA-IIs with PUX10. Furthermore, OLE1 levels were also restored in *mca–II–KOc pux10*, as shown by gel blot analyses and imaging (Supplementary Fig. 23a, b). We should note, however, that *pux10 mca–II–KOc* showed reduced growth at the early stages of development (and the abovementioned sensitivity to tunicamycin) perhaps due to deficiencies in alternative proteostatic pathways like the UPR. Together, these results suggest that efficient ERAD and LDAD in seeds require an MCA–II–dependent proteostatic pathway (Fig. 10b, model).

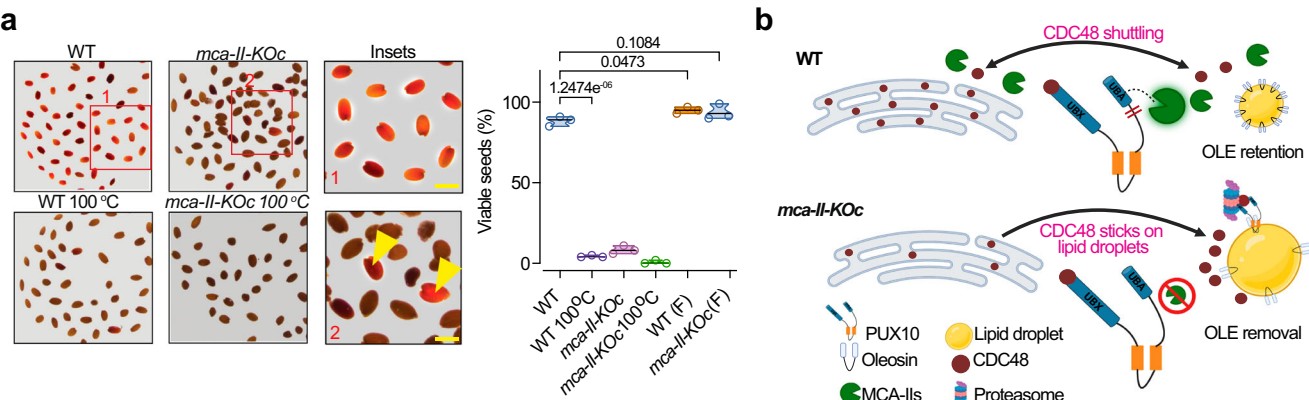

**Fig. 10 | MCA–IIs regulate CDC48 localization and ERAD.** Micrographs from seed viability tests with tetrazolium red staining in WT and *mca-II-KOc*. Seeds treated at 100 °C represent a positive control for dead seeds. Viable seeds (stained red) are highlighted in the insets (yellow arrowheads). Right: quantification of viable seeds. *P*-values were calculated by one-way ANOVA ($N = 3$, $n \geq 176$ seeds in total). **b** A model for the role of MCA-IIs in regulating CDC48 localization to the ER and lipid droplets. In the absence of MCA-IIs, PUX10 is not cleaved and CDC48 is retained on lipid droplets; when MCA-IIs are present, PUX10 is cleaved, releasing CDC48 to localize at the ER. This release is an important step in the spatiotemporal regulation of CDC48 activity and confers seed longevity by modulating ERAD and LDAD. Source data are provided as a Source Data file. Figure 10/panel b Created with BioRender.com released under a Creative Commons Attribution-NonCommercial-NoDerivs 4.0 International license (https://creativecommons.org/licenses/by-nc-nd/4.0/deed.en).

## Discussion

We uncovered a proteolytic pathway modulating spatial proteostatic activity in seeds that involves a regulated balance between ERAD and LDAD. The underlying mechanism involves the specific modulation of the levels of a PUX adapter in the context of seeds (i.e., PUX10). This modulation exerted through the cleavage of PUX10 by MCA-IIs allows CDC48 to shuttle between LDAD and ERAD (i.e., from lipid droplets to the main ER). Interestingly, the deletion of the PUX10 homolog Alveolar soft part locus (ASPL) or Ubiquitin regulatory X4 (UBX4) unlike PUX10 functioning in the ER, impaired ERAD and led to toxicity in mammalian cells and budding yeast, respectively[38,39]. In the seed context, we show that balancing LDAD is an important step in the regulation of ERAD. We assume that this regulation involves other players, and more research is required to uncover missing components and the exact dynamics of PUX10 cleavage. We cannot discount the possibility, that PUX10 is cleaved on lipid droplets still attached to the peripheral ER (en route to their maturation as independent structures), which is supported considering the association of MCAs with both compartments. In line with our results, the localization of UBXD8 (Ubiquitin regulatory X domain-containing protein 8), a mammalian CDC48 adapter that is similar in architecture to PUX10 but no other PUXs, is regulated by a rhomboid pseudoprotease. This interaction is responsible for shuttling CDC48 between the ER and lipid droplets, thereby regulating energy squandering[28,40].

MCAs could potentially function in alternative proteostatic pathways and transiently into the ERAD. We should note the transient nature of MCA–IIs associations with the identified proteins of LDAD studied here (i.e., PUX10, CDC48). This finding fits well in a model whereby small non–stoichiometric complexes form, such as phase–separating condensates. Interestingly, MCAs were recently found to localize to stress-induced condensates[41], and PUXs have been predicted to form condensates[42]. It is thus tempting to speculate that PUXs/MCAs are part of condensates in seeds; we have indeed observed PUX10/MCAs in condensate-like structures. These condensates could enhance the catalytic activity of MCAs by e.g. a local increase of calcium required for the activation of MCAs through the recruitment of calcium–binding proteins. Noteworthy, even very transient interactions between MCAs and their substrates seem sufficient to fulfill proteolytic cleavage events. As was shown in another study, even though MCA–IIs do not associate firmly with the tonoplast, they can still cleave docked peptides there (i.e., PROPEP1)[10].

In non–plant models, suppressing protein anabolism can lead to enhanced longevity; likewise, reduced protein translation via UPR waves (or RIDD) may be functionally equivalent but likely would not suit the maturing seed context (i.e., after-ripening) in which high protein content to nurture the developing embryo is required. Indeed, we found that in older seeds UPR and RIDD are reduced. However, in other contexts, MCA-IIs may be dispensable. On the other hand, in a sensitized environment, where lipid droplet dynamics are altered and ERAD is compromised, MCA-IIs may be important. Proteostatic redundancy has been recognized mainly in animal cells whereby several pathways can converge at the same substrate (e.g., ref. 43). Hence, the lack of MCA-IIs likely activates other pathways that can compensate for their loss, but up to a certain point at which they become indispensable with corresponding phenotypes becoming evident in the long-term (i.e., during seeds storage or senescence). This assumption is consistent with the general lack of (strong) developmental phenotypes for the MCA-II mutant, as it has also been suggested for core ERAD mutants[37], as other pathways come into play.

Finally, we recognize that our study is not exhaustive and parallel pathways could be relevant to the observed phenotypes. For example, we opted for using *bri1-9*, due to the links of MCA-IIs with Ub-proteins. We should note though that Ub is not always linked to ERAD, as has been shown for other substrates like *bri1-5* and the SUBEX-C56Y degradation[37]. Furthermore, lipid droplets in animal cells link to an upregulated ERAD induced by fatty acids, which may suggest a mechanistic explanation for the Mediterranean diet-dependent lifespan extension in humans[44]. In tocopherol-deficient Arabidopsis mutants, oxidized fatty acids accumulate showing decreased seed longevity[45]. It is possible that like *vte2*, alterations in the ERAD/LDAD dynamics cause lipid oxidation which might contribute to the seed longevity phenotype of *mca-II-KOc* mutants. Spatial control of lipid droplet proteins with similar features to oleosins has been reported in yeast for a DOA10 homolog[46]. Hence, the investigation of further links between MCAs and the cytoplasmic ERAD pathway (e.g., DOA10) is justified, including also the type I MCAs. Still, the path linking MCA–II–dependent PUX10 cleavage and CDC48 activity appears to have evolved as an elegant mechanism enabling sustained protein production while maintaining protein homeostasis, - a crucial prerequisite for seed development.

# Methods

## Plant material

All the plant lines used in this study were in the Arabidopsis Columbia-0 (Col-0) ecotype unless stated otherwise. The following mutants were used *smb*[47], *pux10-1* (SAIL_1187), *mca-II-d* (SALK_127688), and *mca-II-f* (GABI_540H06). The *bri1-9* mutant used in this study in Ws genetic background. The two *mca-II* mutants were used as a background for CRISPR. Primers used for genotyping of mutant lines can be found in Supplementary Data 12. The following transgenic lines used in this study were: *PUX10pro:PUX10-GFP*[33] and *MCA-II-fpro:MCA-II-f-GFP*[48]. Seedlings were grown on half-strength Murashige and Skoog (MS) plant agar media under long-day conditions (16h-light/8h-dark, or as indicated) and were harvested, treated, or examined as indicated in the context of each experiment. In all experiments, seedlings, or plants from T1/F1 (co-localization experiments), T2/F2, or T3/4/5 (for physiological experiments) generations were used. Arabidopsis seeds were sterilized and germinated on half-strength MS agar medium under long-day conditions (16 h light/8 h dark). Arabidopsis plants for crosses, phenotyping of the above-ground part, and seed collection were grown on soil in a plant Aralab chamber at 22 °C/19° and a light intensity of 150 μmol m$^{-2}$ s$^{-1}$ with 60% relative humidity stabilized by an infrared sensor. The seeds harvested at the same time under the same condition were used for experiments unless otherwise indicated in the text or figure legends. *Nicotiana benthamiana* plants were grown in Aralab or Percival cabinets at 22 °C, 16-h-light/8-h-dark cycles, and a light intensity of 150 μmol m$^{-2}$ s$^{-1}$.

## Metacaspase phylogeny

Alignments of MCA sequences were performed in MUSCLE. Unrooted trees were constructed using the neighbor-joining method using the yeast homolog as an outgroup. A phylodendrogram was constructed using MEGA11 and PAUP software (http://paup.csit.fsu.edu). The bootstrap analysis was performed with 1,000 repeats, and branches with bootstrap values over 70% were retained. The protein sequences used can be found in Supplementary Data 11.

## Drugs and stainings

The stock solutions of 2 mM FM4-64, 10 mM Tunicamycin, 40 mM Kifunensin, 1 mM thapsigargin, 10 mM CB-5083 (1-(4-(benzylamino)-7,8-dihydro-5H-pyrano-(4,3-d)-pyrimidin-2-yl)-2-methyl-1H-indole-4-carboxamide), 1 mM BCECF (2',7'-Bis-(2-Carboxyethyl)-5-(and-6)-Carboxyfluorescein, Acetoxymethyl, Ester), LipidTOX (1:1,000 dilution, Thermo, H34477), 50 mM MG132, 2 mM Concanamycin A (Con A), 33 mM wortmannin (Wm) and 10 mM E64d (Thermo) were dissolved in dimethyl sulfoxide (DMSO). Propidium iodide (PI) and cycloheximide (CHX) were dissolved in water. These inhibitors, drugs, and dyes were diluted in a half-strength MS medium with corresponding concentration and duration, and the final DMSO concentration was ≤ 0.1% (v/v) in all experiments. For kifunensin and thapsigargin (due to their instability), seedlings were transferred from drug-free plates to plates containing the corresponding drugs for 3 days, scoring phenotypes each day. For short-term treatments, vertically grown 4–5 day-old Arabidopsis seedlings (or as indicated) were incubated in a half-strength liquid MS medium containing the corresponding compound for each specific time course treatment. For confocal microscopy of reporters and mutants, fluorescence images were captured on Leica SP8 or Zeiss780 microscopes and were processed with ImageJ (National Institutes of Health). Cell contours were visualized with propidium iodide (PI) (Molecular Probes) or FM4-64. For the FDA-PI viability staining, seedlings were mounted on a glass slide in FDA solution (1 μL dissolved FDA stock solution [2 mg in 1 ml acetone] in 1 ml of 1/2 MS) supplemented with 10 μg/ml PI. For the seed viability test, tetrazolium red assays used[49]. In short, dry seeds from different genotypes were incubated in the dark in an aqueous solution of 1% (w/v) 2,3,5-triphenyl tetrazolium (TZ) at 28 °C for 24–48 h with or without indicated treatment. Seeds were rinsed in water before imaging. For lipid droplet staining, LipidTOX at a 500-fold dilution was used.

## Controlled deterioration treatment (CDT) assay

For CDT, seeds were incubated at 37ºC for 28 h in water for high humidity conditions. After 28 h, seeds were surface sterilized in 12% (v/v) sodium hypochlorite solution (commercial bleach) for 15 min followed by 1 wash with 70% (v/v) ethanol, 3 washes with ddH$_2$O, and O/N incubation at 4 °C for vernalization. Vernalized seeds were sown on ½ MS petri plates under sterile conditions and their germination rate was monitored.

## Protease activity assay

Protease activities were measured by fluorogenic peptide-based substrate (EGR-AMC: H-Glu-Gly-Arg-7-amino-4-methylcoumarin)[50]. Plant (0.01 g seeds or equal amount of 7 day-old seedlings) extract from WT and different mutants using reaction buffer: 50 mM HEPES, pH 7.4, 0.1 % (w/v) 3-[(3-cholamido- propyl) dimethylammonio]-1-propanesulfonate (CHAPS), 50 mM CaCl$_2$, 5 mM dithiothreitol (DTT); EGR-AMC were added at 50 μM final. The release of AMC was measured every 2 min at 30 °C with a Microtiter Plate Fluorometer (Microplate reader Fluostar Omega) using an excitation wavelength of 360 nm and an emission wavelength of 460 nm. Data time points were analyzed by the Omega Fluostar software, and activities were expressed in fluorescence units/min/mg or μg of total protein. Protein concentration was determined using the Bradford reagent (Bio-Rad).

## Clonings and production of transgenic lines

Electrocompetent Agrobacterium (*Agrobacterium tumefaciens*) strain C58C1 Rif$^R$ (pMP90) or GV3101 Rif$^R$ (i.e., a cured nopaline strain commonly used for infiltration) was used for electroporation, *N. benthamiana* infiltration and floral dip transformation in Arabidopsis. The following constructs used in the study were: *pGWB4-PUX10pro:-PUX10-GFP*, *pGWB16-PUX10pro:PUX10-myc*, *PMDC140-PUX10pro:-PUX10-mCherry*, *35Spro:mCherry-PUX10*, *35Spro:mCherry-CDC48a*[33]. Transcriptional and translational reporters and overexpression constructs used in this study were produced through either GATEWAY (Thermo Fisher Scientific) cloning using pENTR/D and pENTR5' vectors or through GOLDENGATE (Addgene) in the following backbones: (i) pGWB505 and pGWB560[51] (ii) pMDC32[52] (iii) (iv) pICSL86900 and pICSL86922 (Addgene). The cDNA of *MCA-II-a* was PCR amplified with Phusion™ High-Fidelity DNA Polymerase & dNTP Mix (Thermo Fisher Scientific, F530N) using cDNA from 7 day-old seedlings grown under standard conditions (see "plant material"). Vector collection for split-GFP assays was described in ref. 22 and the relevant GFP11-MCA-II-a construct was prepared by GATEWAY cloning. For BiFC assays, all genes of interest were amplified without a stop codon, fused to C-terminal either with nVenus or cCFP in the destination vector pICH86988 unless otherwise stated (plasmids: pCR8GW-TOPOcCFP for C-terminal tagging and pCR8GW-TOPOnVENUS). *MCA-II-a$^{PD}$* was generated with site-directed mutagenesis (Stratagene, Lightning) using the pENTR of *MCA-II-a* or *MCA-II-b* and inserting the C136A substitution for MCA-II-a and the C139A for MCA-II-b. Constructs of *MCA-II-a/b* or *MCA-II-a/b$^{PD}$* were generated by GATEWAY cloning with pENTR into different destination vectors which have different tags in N- or C-termini. The BRI1/Bri1-9-GFP lines were generated by transforming the constructs *pGWB601-RPS5apro:BRI1/bri1-9-GFP* in WT and *mca-II-KOc* backgrounds. The coding sequence of *CDC48a, PUX10* were PCR amplified with Phusion™ High-Fidelity DNA Polymerase & dNTP Mix (Thermo Fisher Scientific, F530N) using the cDNA from 7 day-old seedling with

pENTR™/D-TOPO™ Cloning Kit (ThermoFisher Scientific). Primer sequences used for the amplification of promoters and genes are listed in Supplementary Data 12.

### Construction of MCA-II mutants

The pICSL binary vector series was utilized to generate CRISPR lines in this study. Primer sequences used for the amplification of promoters and genes are listed in Supplementary Data 12. To generate the Cas9 expression cassettes, the RPS5a and Cas9z coding sequences and the E9 terminator were amplified using primers flanked with BpiI restriction sites associated with Golden Gate compatible overhangs (Supplementary Data 12). Combinations of three Level 0 vectors containing respectively a promoter, a Cas9z coding sequence and a terminator were assembled in Level 1 vector pICH47811 (Position 2, reverse) by the same 'Golden Gate' protocol but using 0.5 μl of BpiI enzyme (10U/μl, ThermoFisher) instead of 0.5 μl of BsaI-HF. To create the *MCA-IIs* deletion mutant, a multiplexed editing approach was used. The sgRNAs were designed using the CRISPR-P 2.0 (http://crispr.hzau.edu.cn/CRISPR2) and CHOP-CHOP (https://chopchop.cbu.uib.no/). To generate the sgRNA expression cassettes, DNA fragments containing the classic or the 'EF' backbone with 7, 67 or 192 bp of the U6-26 terminator were amplified using primers flanked with BsaI restriction sites associated with GOLDENGATE compatible overhangs (Supplementary Data 12). The amplicons were assembled with the U6-26 promoter (pICSL90002) in Level 1 vector pICH7751 (gRNA-MCA-II-e, Position 3), pICH7761 (gRNA-MCA-II-b, Position 4), pICH7772 (gRNA-MCA-II-a, Position 5) and pICH7781 (gRNA-MCA-II-c, Position 6) by the 'GOLDENGATE' protocol using the BsaI-HF enzyme. Combinations of three Level 1 vectors containing a red seed coat maker (FAST-Red, pICSL11015, Position2, *OLE1pro:OLE1-RFP*), a Cas9 expression cassette, and four sgRNA expression cassettes were assembled in Level 2 pAGM4723 (without an overdrive) or pICSL4723 (with an overdrive) by the 'GOLDENGATE' protocol using the BpiI enzyme. A new level 1 vector pICH7751 (gRNA-MCA-II-d-1, Position 3), pICH7761 (gRNA-MCA-II-d-2, Position 4) were assembled for regenerating knock-out line of MCA-II-d by the 'GOLDENGATE' protocol using the BsaI-HF enzyme and the level 2 pAGM4723 were assembled with the combination of level 1 vectors containing a green seed coat maker (FAST-Green, pICSL11042, Position2, *OLE1pro:OLE1-GFP*), a Cas9 expression cassette, two sgRNA expression cassettes and linker "pele 5" by the 'GOLDENGATE' method using the BpiI enzyme. All the plasmids were purified using a ThermoFisher Scientific kit on *Escherichia coli* DH10B electrocompetent cells selected with appropriate antibiotics and X-gal. All the plasmid identification numbers refer to those denoted in the 'Addgene database' (www.addgene.org/). We selected red fluorescing seeds and screened the resulting seedlings for first-round mutation. CRISPR clean lines were selected based on the crossing to WT and to get the segregation lines for further screening of non-red seed coat and resequencing. Then we selected green seeds and screened for additional MCA-II-d mutation lines with sequencing. CRIPSR clean lines were selected based on the backcrossing to WT and further screening of the non-green seed coat and resequencing.

### RNA extraction, RNA-seq and quantitative RT-PCR analysis

Total RNA from the seedlings was extracted using RNeasy Plant Mini Kit with DNaseI digestion (QIAGEN). Reverse transcription was carried out with 500 ng of total RNA using the iScript cDNA synthesis kit (Bio-Rad) according to the manufacturer's protocol. Quantitative PCR with gene-specific primers was performed with the SsoAdvanced SYBR Green Supermix (Bio-Rad) on a CFX96 Real-Time PCR detection system (BioRad). Signals were normalized to the reference genes ACTIN7 using the DCT method and the relative expression of a target gene was calculated from the ratio of test samples to WT. Primer sequences used for the amplification of promoters and genes are listed in Supplementary Data 12. For each genotype, two biological replicates were assayed in three qPCR replicates. qRT-PCR primers were designed using QuantPrime. For RNA-seq the concentration of RNA was determined by Qubit® RNA HS Assay Kit (New England BioNordika BioLab). All the RNA samples were treated with DNase I (Thermo Fisher Scientific) and further enriched with NEBNext® Poly(A) mRNA Magnetic Isolation Module (ThermoFisher Scientific). The RNA was measured with Qubit® RNA HS Assay Kits again and libraries were prepared with NEBNext® Ultra™ II RNA Library Prep with Sample Purification Beads (Invitrogen Life Technologies) (Ambion Applied Biosystem) and NEBNext® Multiplex Oligos for Illumina® (Dual Index Primers Set 1) (New England BioNordika BioLab). cDNA library quality was monitored with Agilent DNA 7500 Kit (Agilent Technologies Sweden AB). cDNA libraries were sequenced with a paired-end sequencing strategy to produce $2 \times 150$-bp reads using Novogen sequencers and 20 million reads per sample (Novogene, England).

### Cell fractionation

Cell fractionation was done using *MCA-II-apro:MCA-II-a-GFP* lines based on sucrose gradient density ultracentrifugation as in ref. 21. More specifically, seedlings (5 g) of the above line were ground followed by the addition of homogenization buffer (50 mM Tris HCl [pH 8.2], 2 mM EDTA pH 8.0, 1 mM DTT and protease inhibitors cocktail from Sigma-Aldrich at a 1:100 dilution plus 1 mM phenylmethylsulfonyl fluoride [PMSF]). The extract was filtered through miracloth and centrifuged at 5000 x g for 5 min for the removal of organelles and tissue debris, followed by ultracentrifuge at 100,000 x g for 45 min and the pellet (microsomes) was resuspended in 1 ml resuspension buffer (25 mM Tris-HCl [pH 7.5]), 10% sucrose, 1 mM PMSF, 2 mM EDTA, 1 mM DTT and protease inhibitors cocktail from Sigma-Aldrich at a 1:100 dilution. The resuspended microsomes were loaded in an 8 ml 20-50 (v/v) % sucrose gradient in 10 mM Tris-HCl [pH 7.5], 2 mM EDTA, 1 mM DTT and 0.1 mM PMSF. Samples were ultracentrifuged overnight at 100,000 x g. After centrifugation, subcellular fractions were collected in 1.5 ml Eppendorf and processed for immunoblot with the following antibodies: α-GFP (Santa Cruz Biotechnology), α-BIP2 (Agrisera), and α-APX1 (Agrisera). For the $Mg^{2+}$ condition, 5 mM $MgCl_2$ were added to the homogenization, resuspension, and ultracentrifuge buffers.

### Protease protection assay and MCA-II-a binding on microsomal fraction

The protease protection assay was done using microsomes purified from cell fractionation. WT seedlings (1 g) were ground followed by the addition of homogenization buffer (50 mM Tris HCl [pH 8.2], 2 mM EDTA pH 8.0, 1 mM DTT, and protease inhibitors cocktail from Sigma-Aldrich at a 1:100 dilution plus 1 mM PMSF). The extract was filtered through miracloth and centrifuged at 5000 x g for 5 min for the removal of organelles and tissue debris, followed by ultracentrifuge at 100,000 x g for 45 min and the pellet (microsomes) was incubated with 1 μM of Protease K (Thermo scientific) for 15 min at 47 °C. To release microsome lumen content, the microsomes were treated with 1% (v/v) Triton X-100.

### Fatty acid analysis

The total fatty acid content and composition of mature seeds were determined by direct transmethylation followed by gas chromatography with a flame ionization detection as in ref. 53.

### BIFC and Split-GFP assays

*Nicotiana benthamiana* leaves were infiltrated with the Agrobacterium strain GV3101 at $OD_{600} = 0.5$ carrying the constructs, as indicated in the corresponding figures. After infiltration, leaf discs from infiltrated sports were imaged 4 days post-infiltration. All samples were imaged with a 40x oil objective ($NA = 1.3$) of a Leica SP8 confocal microscope. GFP was excited using a 488 nm argon laser and detected at 500-530 nm. YFP was excited using a 514 nm argon laser and emission

was collected at 520-550 nm, while chlorophylls were detected between 653 and 676 nm and excited at 561 nm.

## Immunocytochemistry, PLA, and imaging

Immunocytochemistry was done as described previously (ref. 54). The primary antibodies used were rabbit α-MCA-II-a (1:500, custom against the recombinant MCA-II-a), α-BIP2 (1:500) and goat α-CDC48a (VCP1) (1:500, Abcam. 206320). In brief, samples were incubated with primary antibody at 4 °C overnight and washed three times with PBS-T, and then incubated for 90 min with Alexa Fluor® 488 AffiniPure Donkey α-Goat IgG (H + L) secondary antibody (Jackson ImmunoResearch, 705-545-147) diluted 1:200-250, After washing in PBS-T and incubating with DAPI (1 µg/mL), specimens were mounted in Vectashield (Vector Laboratories) medium and observed within 48 h. PLA immunolocalization was done as described previously[55]. Primary antibody combinations diluted 1:200 goat α-CDC48a, 1:200 for α-GFP mouse (Sigma-Aldrich, SAB2702197), 1:200 for α-FLAG mouse (Sigma-Aldrich, F1804), 1:200 for α-RFP mouse (Agrisera, AS15 3028) and 1:200 for α-GFP rabbit (Millipore, AB10145) were used for overnight incubation at 4 °C. Roots were then washed with microtubule-stabilizing buffer (MTSB: 50 mM PIPES, 5 mM EGTA, 2 mM MgSO₄, 0.1% [v/v] Triton X-100) and incubated at 37 °C for 3 h either with α-mouse plus and α-rabbit minus, or α-mouse plus and α-goat minus for PLA assay (681 Duolink, Sigma-Aldrich). PLA samples were then washed with MTSB and incubated for 3 h at 37 °C with ligase solution as described in ref. 56. Roots were then washed 2x with buffer A (Sigma-Aldrich, Duolink) and treated for 4 h at 37 °C in a polymerase solution containing fluorescent nucleotides as described (Sigma-Aldrich, Duolink). Samples were then washed 2x with buffer B (Sigma-Aldrich, Duolink), with 1% (v/v) buffer B for another 5 min, and then the specimens were mounted in Vectashield (Vector Laboratories) medium for confocal imaging.

## Quantification of fluorescent intensity

To create the most comparable lines to measure the fluorescence intensity of reporters in multiple mutant backgrounds, we crossed homozygous mutant bearing the marker with either a WT plant (outcross to yield progeny heterozygous for the recessive mutant alleles and the reporter) or crossed to a mutant only plant (backcross to yield progeny homozygous for the recessive mutant alleles and heterozygous for the reporter). Fluorescence was measured as the mean gray value with subtraction of the background. GFP and chloroplast autofluorescence was excited with the 488 nm line of an argon laser, mCherry and RFP with a 561 nm diode laser, YFP with the 514 nm line of an argon laser, and LipidTOX Deep Red with a 633-nm helium/neon laser. Fluorescence emission was detected between 495 and 510 nm for GFP, 522-550 nm for YFP, 600-625 nm for mCherry and RFP, and 637-650 nm for LipidTOX Deep Red. Chloroplast autofluorescence was imaged between 670 and 700 nm. For multilabeling studies, detection was performed in a sequential line-scanning mode. The apparent diameter of LDs observed by CLSM was estimated using Fiji software (https://fiji.sc/) by manually drawing the diameter using the "line" tool and measuring it with the "measure" function of the software. For BiFC excitation wavelengths and emission, filters were 514 nm/band-pass 530-550 nm for YFP, 561 nm/band-pass 600-630 nm for RFP and 488 nm/band-pass 650-710 nm for chloroplast autofluorescence. The objective used was an HC PL APO 40x/1,30 oil CS2 with NA = 1.3 (Leica SP8 confocal system).

## Ubiquitin cleavage assay

Recombinant proteins of GST-PROPEP1 (metacaspase substrate), MCA-II-a, MCA-II-aPD, and Usp2-cc were purified as in ref. 10, and the assays were done with the protocols described in ref. 57. The deubiquitylating activity of Usp2-cc, MCA-II-a, and MCA-II-f was assayed against recombinant human K48-linked tetra-ubiquitin

(BostonBiochem, #UC-210B) by incubating the purified proteases with 2 µg substrate at 37 °C for 30 min in either MC-II-f reaction buffer (50 mM MES, pH 5.5, 150 mM NaCl, 10% sucrose, 0.1% CHAPS, 10 mM DTT) or MCA-II-a reaction buffer (50 mM Hepes, pH 7.5, 150 mM NaCl, 10% glycerol, 50 mM CaCl₂, 10 mM DTT). Reactions were terminated by adding Sodium dodecyl-sulfate polyacrylamide gel electrophoresis (SDS-PAGE) sample buffer with an additional 50 mM EGTA for MCA-II-a reaction buffer, subjected to SDS-PAGE (12% polyacrylamide) and subsequent silver staining (Thermo Fisher Scientific).

## Immunoblotting

In general, samples were flash-frozen in liquid N₂ and kept at -80 °C until further processing. The samples were crushed using a liquid N₂-cooled mortar and pestle, and the crushed material was transferred to a 1.5-ml or 15-ml tube. Extraction buffer (EB; 50 mM Tris-HCl pH 7.5, 150 mM NaCl, 10% [v/v] glycerol, 2 mM ethylenediamine tetraacetic acid [EDTA], 5 mM DTT, 1 mM PMSF, Protease Inhibitor Cocktail [Sigma-Aldrich] and 0.5 % [v/v] IGEPAL CA-630 [Sigma-Aldrich]) was added according to the plant material used. The lysates were pre-cleared by centrifugation at 16,000 g at 4 °C for 15 min, and the supernatant was transferred to a 1.5-ml tube. This step was repeated two times, and the protein concentration was determined by the RC DC Protein Assay Kit II (Bio-Rad). 2X Laemmli buffer was added, and proteins were separated by SDS-PAGE (1.0 mm thick 4-12% [w/v] gradient polyacrylamide Criterion Bio-Rad) in 3-(N-Morpholino) propane sulfonic acid (MOPS) buffer (Bio-Rad) at 150 V or native polyacrylamide gel (Bio-Rad, Any kD TGX gels). Subsequently, proteins were transferred onto polyvinylidene fluoride (PVDF; Bio-Rad) membrane with 0.22-µm pore size. The membrane was blocked with 3% (w/v) BSA fraction V (Thermo Fisher Scientific) in phosphate buffered saline-Tween 20 (PBS-T) for 1 h at room temperature (RT), followed by incubation with horseradish peroxidase (HRP)-conjugated primary antibody at RT for 2 h (or primary antibody at RT for 2 h and corresponding secondary antibody at RT for 2 h). The following antibodies were used: rabbit α-OLE1 (anti-rS3)[33], rat α-tubulin (Santa Cruz Biotechnology, 1:1000), rabbit α-GFP (Millipore, AB10145, 1:10,000), mouse α-RFP (Agrisera, AS15 3028, 1:5000), rabbit α-BIP2 (Agrisera, 1:2000), rabbit α-UBQ11 (Agrisera, 1:2000), rabbit α-MCA-II-a (1:1000), goat α-CDC48a (VCP1) (diluted 1:2000, Abcam. 206320), rabbit α-PBA1 (Agrisera, 1:2000), rabbit α-PAG1 (Agrisera, 1:2000), rabbit α-BIP (Agrisera, 1:2000), rabbit α-ACTIN (Agrisera, 1:2000), α-mouse (Amersham ECL Mouse IgG, HRP-linked whole Ab [from sheep], NA931, 1:10,000), α-rabbit (Amersham ECL Rabbit IgG, HRP-linked whole Ab [from donkey], NA934, 1:10,000), α-rat (IRDye® 800 CW Goat anti-Rat IgG [H + L], LI-COR, 925-32219, 1:10,000) and α-rabbit (IRDye ® 800 CW Goat anti-Rabbit IgG, LI-COR, 926-3221, 1:10,000). Chemiluminescence was detected with the ECL Prime Western Blotting Detection Reagent (Cytiva, GERPN2232) and SuperSignal™ West Femto Maximum Sensitivity Substrate (Thermo Fisher Scientific). The bands were visualized using an Odyssey infrared imaging system (LI-COR).

## Total proteome analysis, DIA-MS, COFRADIC, and TAP-MS

An equal number of seeds of WT and mca-II-KO were used for the analysis of the total proteome using LC-MS/MS. For COFRADIC, a shortened protocol was used as previously with the following modifications[48]. To achieve a total protein content of 1 mg, 0.2 g frozen ground tissue was resuspended in 1 mL of buffer containing 1% (w/v) 3-[3-cholamidopropyl)-dimethylammonio]-1-propane sulfonate (CHAPS), 0.5% (w/v) deoxycholate, 5 mM ethylenediaminetetraacetic acid, and 10% glycerol in 50 mM HEPES buffer, pH 7.5, further containing the suggested amount of protease inhibitors (one tablet/10 mL buffer) according to the manufacturer's instructions (Roche Applied Science). The sample was centrifuged at 16,000 g for 10 min at 4 °C, and guanidinium hydrochloride was added to the cleared supernatant

to reach a final concentration of 4 M. Protein concentrations were measured with the DC protein assay (Bio-Rad), and protein extracts were further modified for N-terminal COFRADIC analysis as described in ref. 58. Col-0 (WT) primary amines were labeled with the N-hydroxysuccinimide (NHS) ester of $^{12}C_4$-butyrate and mutants with NHS-$^{13}C_4$-butyrate, resulting in a mass difference of ~4 Da between light ($^{12}C_4$) and heavy ($^{13}C_4$) labeled peptides. After equal amounts of the labeled proteomes had been mixed, tryptic digestion generated internal, non-N-terminal peptides that were removed by strong cation exchange at a low pH[58]. Due to the low amount of input material, the COFRADIC protocol was cut short after the first reversed-phase-high performance liquid chromatography (RP-HPLC) step, and the resulting 15 fractions were subjected immediately for identification by LC-MS/MS. For TAP-MS experiments, four to five weeks-old Arabidopsis transgenic plants expressing MCA-II-a-TAPa, MCA-II-a$^{PD}$-TAPa, MCA-II-b-TAPa, MCA-II-b$^{PD}$-TAPa and sGFP-TAPa were harvested (2-4 g, fresh weight) and ground in liquid $N_2$ in 2 volumes of extraction buffer (50 mM Tris-HCl pH 7.5, 150 mM NaCl, 10% glycerol, 0.1% Nonidet P-40 and 1× protease inhibitor cocktail; Sigma-Aldrich, 1:100 dilution). The analyses and further processing were done as described in ref. 59. For DIA (data independent acquisition), samples were run into SDS-PAGE as gel plugs. Each gel band was subjected to reduction with 10 mM DTT for 30 min at 60 °C, alkylation with 20 mM iodoacetamide for 45 min at room temperature in the dark and digestion with trypsin (sequencing grade, Thermo Fisher Fischer Scientific, 90058), and incubated over-night at 37 °C. Peptides were extracted twice with 5% formic acid, 60% acetonitrile and dried under vacuum. Samples were analyzed by LC-MS using Nano LC-MS/MS (Dionex Ultimate 3000 RLSCnano System, Thermofisher) interfaced with Eclipse (Thermofisher). Samples were loaded on to a fused silica trap column Acclaim PepMap 100, 75 μm x 2 cm (Thermo Fisher Scientific). After washing for 5 min at 5 μl/min with 0.1% TFA, the trap column was brought in-line with an analytical column (Nanoease MZ peptide BEH C18, 130 A, 1.7 μm, 75 μm x 250 mm, Waters) for LC-MS/MS. Peptides were fractionated at 300 nl/min using a segmented linear gradient 4-15% B in 30 min (where A: 0.2% formic acid, and B: 0.16% formic acid, 80% acetonitrile), 15-25% B in 40 min, 25-50% B in 44 min, and 50-90% B in 11 min. Solution B then returns at 4% for 5 min for the next run. DIA workflow was used to analyze the eluted peptides. MS scan range were set at 400-1200, resolution 12,000 with AGC set at 3E6 and ion time set as auto. 8 m/z window were set to sequentially isolate (AGC 4E5 and ion time set at auto) and fragment the ions in C-trap with relative collision energy of 30. The MSMS were recorded with Resolution of 30,000. Raw data were analyzed with predicted library from uniprot *Arabidopsis thaliana* reference proteome fasta database for library-free search using DIA NN 1.8.1 (1) with recommended settings. The results were filtered for both PEP (an estimate of the posterior error probability for the precursor identification, based on scoring with neural networks) filter <0.01 and PG.Q (Protein Group Q Value) filter <0.01. Protein group MaxLFQ value were used for group comparisons.

### Visualization of networks and analyses

Cytoscape v. 3.5.1 was used. Tab-delimited files containing the input data were uploaded. Unless otherwise indicated, the default layout was an edge-weighted spring-embedded layout, with NormSpec used as edge weight. Nodes were manually re-arranged from this layout to increase visibility and highlight specific proximity interactions. The layout was exported as a PDF and eventually converted to a.TIFF file with Lempel-Ziv-Welch (common name LZW) compression.

### Statistics and reproducibility

Graphs were generated by GraphPad Prism v. 10.2.0. Pearson or Spearman correlation coefficients on images were calculated via Fiji, coloc2 tool or manually and plotted in GraphPad Prism. When relevant, the correlation confidence intervals were also calculated and represented. All statistical data show the mean ± s.d. (box plots) or the distribution of values (violin plots, kernal density) of at least three biologically independent experiments or samples, or as otherwise stated. Individual data points are on the plots. For violin plots, datasets were smoothed using heavy smoothing which gives a better idea of the overall distribution. In captions, *N* denotes biological replicates, and "*n*" technical replicates or population size (or as indicated). Each data set was tested whether it followed normal distribution when $N \geq 3$ by using the Shapiro normality test integrated into the GraphPad Prism. The significance threshold was set at $P < 0.05$, and the calculated *P*-values are shown in the graphs or as otherwise stated. For *P*-values below 0.001, the value was calculated using Microsoft EXCEL 365. Details of the statistical tests applied, including the choice of the statistical method, are indicated in the corresponding figure legend. In boxplots or violin plots, upper and lower box boundaries, or lines in the violin plots when visible, represent the first and third quantiles, respectively, horizontal lines mark the median and whiskers mark the highest and lowest values.

### Reporting summary

Further information on research design is available in the Nature Portfolio Reporting Summary linked to this article.

## Data availability

All data, code, and materials used in the analysis are available and were deposited in public repositories. Gene sequence information of MCAs is provided in Supplementary Data 1 and 12. The mass spectrometry proteomics data have been deposited to the ProteomeXchange Consortium via the PRIDE partner repository with the dataset identifiers PXD048890 and PXD049287. The processed data are available online at https://zenodo.org/records/12684164. The raw RNAseq data of this article have been deposited to the BioStudies database with the dataset accession number S-BSST1310. Source data are provided with this paper.

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

## Acknowledgements
We acknowledge Cyril Zipfel (The Sainsbury Laboratory), Ikuko Hara–Nishimura (Kyoto University), Hans Thordal–Christensen, and Hannele Tuominen (Swedish University of Agricultural Sciences) for sharing materials. This work received funding from the Knut and Alice Wallenberg Foundation 2018.0026 (PVB, PNM); EPIC–XS, Horizon 2020 programme of the European Union project number 823839 (KG, PNM); European Research Council (ERC) Starting Grant 'R–ELEVATION' 101039824 (P Ding); European Research Council (ERC) Consolidator Grant PLANTEX, 101126019, 10.3030/101126019 (PNM); Future Leader Fellowship from the Biotechnology and Biological Sciences Research Council (BBSRC) BB/R012172/1 (P Ding); European Union and Greek national funds through the Operational Program Competitiveness, Entrepreneurship, and Innovation, T2EΔK–00597 under the call RESEARCH–CREATE–INNOVATE "BIOME" (PNM); European Union Horizon 2020 Marie Curie–RISE Grant PANTHEON 872969 (PNM); Hellenic Foundation of Research and Innovation Grant–Theodoros Papazoglou–Always Strive for Excellence NESTOR 1426 (PNM); Hellenic Foundation of Research and Innovation Fellowship 5947 (IHH); The Swedish Research Council (VR) 2019–04250 (PVB, PNM); Carl Trygger Foundation (CTS) 22:2025 (PVB); European Union Marie Skłodowska–Curie Action IF project 656011 (PNM); DESTINY (BOF–UGent) (SS, FVB). Views and opinions expressed are those of the author(s) only and do not necessarily reflect those of the European Union or the European Research Council Executive Agency. Neither the European Union nor the granting authority can be held responsible for them.

## Author contributions
Conceptualization: PNM, PVB, JDGJ, CL; Methodology: CL, IHH, TP, SHR, EAM, AM, EP, SS, EGB, P Dörmann, SDA, KG, FRC, P Ding, MN, FVB; Investigation: CL, IHH, PNM; Visualization: CL, IHH, PNM; Funding acquisition: PVB, PNM, FVB, P Ding, KG; Project administration: PNM; Supervision: PNM; Writing – original draft: PNM, CL, IHH, PVB; Writing – review & editing: all authors.

## Funding

## Competing interests
The authors declare no competing interests.

## Additional information

[1]State Key Laboratory of Biocontrol, Guangdong Key Laboratory of Plant Resources, School of Life Sciences, Sun Yat–Sen University, 510275 Guangzhou, China. [2]Department of Biology, University of Crete, 71500 Heraklion, Greece. [3]Institute of Molecular Biology and Biotechnology, Foundation for Research and Technology—Hellas, 71500 Heraklion, Greece. [4]Department of Plant Biology, Uppsala BioCenter, Swedish University of Agricultural Sciences and Linnean Center for Plant Biology, 75007 Uppsala, Sweden. [5]Department of Medicine, Health and Medical University, 14471 Potsdam, Germany. [6]Plant Ecology and Evolution, Department of Ecology and Genetics, Evolutionary Biology Centre and the Linnean Centre for Plant Biology in Uppsala, Uppsala University, 75236 Uppsala, Sweden. [7]Department of Molecular Sciences, Uppsala BioCenter, Swedish University of Agricultural Sciences and Linnean Center for Plant Biology, 75007 Uppsala, Sweden. [8]VIB–Ugent Center for Plant Systems Biology, Technologiepark 71, 9052 Ghent, Belgium. [9]Department of Plant Biotechnology and Bioinformatics, Ghent University, Technologiepark 71, 9052 Ghent, Belgium. [10]Instituto de Bioquımica Vegetal y Fotosıntesis, Consejo Superior de Investigaciones Cientıficas (CSIC)–Universidad de Sevilla, 41092 Sevilla, Spain. [11]Departamento de Bioquımica Vegetal y Biologıa Molecular, Facultad de Biologıa, Universidad de Sevilla, 41012 Sevilla, Spain. [12]University of Bonn, Institute of Molecular Physiology and Biotechnology of Plants (IMBIO), Karlrobert Kreiten Straße 13, 53115 Bonn, Germany. [13]Université Paris–Saclay, INRAE, AgroParisTech, Institut Jean–Pierre Bourgin (IJPB), 78000 Versailles, France. [14]VIB Center for Medical Biotechnology, Technologiepark-Zwijnaarde 75, B9052 Ghent, Belgium. [15]Department of Biomolecular Medicine, Ghent University, Technologiepark-Zwijnaarde 75, B9052 Ghent, Belgium. [16]Institute of Biology Leiden, Leiden University, 2333 BE Leiden, The Netherlands. [17]The Sainsbury Laboratory, University of East Anglia, Colney Lane, NR47UH Norwich, UK. [18]These authors contributed equally: Chen Liu, Ioannis H. Hatzianestis. ✉e-mail: Panagiotis.moschou@uoc.gr

