## [Peer Review File · Nature Communications]

REVIEWER COMMENTS

Reviewer #1 (Remarks to the Author):

The manuscript is a beyond reproach and makes great contributions to the understanding of the roles of MCA-II genes. The unravelling of the relationships between these genes and other components of PCD systems and protein synthesis/degradation in seeds is a trove.

As a reviewer I made a few "ridiculous" corrections in the .pdf file. As a researcher, I was curious to know if the mature seeds of Arabidopsis used in certain assays (such as protease activity assays) were peeled. This curiosity is born from recent reports of PCD events in seed coats of soybeans and Vigna beans. As the authors evaluated the involvement of MCD-II genes on developmental PCD processes only in root cap systems, I wondered if this would be so also in the case of seed coats. But that's just something to be think about.

Other than that, I congratulate the authors on such a refined piece of work.

Reviewer #2 (Remarks to the Author):

In plants, there is still a lot to learn about factors regulating seed germination and seed longevity. Seed longevity in storage is an important trait for crop seed production and for seed banks. Here the authors make an important contribution to this research area. The role of metacaspases in the above processes has been totally overlooked because of a problem of genetic redundancy that could not be resolved without the availability of metacaspase multi-KO lines. The authors are the first to have dedicated time to creating multi-KO lines for all the MC II genes, and this unlocked a raft of new observations.

This is a well-written Ms, with experiments beautifully carried out and displayed. The Ms is an important study that combines and integrates many different approaches. The work brings a key cellular insight into seed longevity. In addition, the data sets generated give several new leads for future studies, which will establish a better understanding of key cellular processes which integrate metacaspases. The new cellular regulation involving a protease that is identified here will be of interest to all cell biologists.

There are only a few small issues to resolve, which are listed below. And possibly, the hypothesis to explain the loss of seed viability in the mutant background needs to be refined (point 12).

General comment:

Why is the MC numbering convention different from the one in TAIR? For example, here, AtMC4 (At1g79340) is AtMC-II-a, and in TAIR, it is AtMC-II-d. This brings confusion when looking at the

literature. As far as I can tell, Ref 5 only indicates the format of the metacaspase names without specifics. Should the authors follow the TAIR annotation?

AtMCA-II-a is systematically tested in various experiments, but there is no rationale to explain why various other metacaspases are tested simultaneously: sometimes b or sometimes d. Could this be explained?

Specific comments:

1. Abstract: wrong use of the word dormancy in this context. See Bewley, J.D., 1997. Seed germination and dormancy. *The plant cell*, 9(7), p.1055. I would think it should be quiescence, not dormancy.
2. Figure 2 line 665. 'the proteasomal subunits (GO, protein folding) such as PAG1 (20S proteasome alpha subunit G- 1: 1.3-fold) and PBA1 (20S proteasome subunit beta 1: not found in WT), were verified using α -PAG1 and α -PBA1 antibodies in Fig S5G'. Looking at Fig S5G, it is PAG1 which is absent of WT, present in the mutant, not PBA1, which has the same pattern in WT and mutants.
3. Line 130 'To determine the localisations of MCA-II-a and MCA-II- d, we generated transgenic plants carrying a translational fusion construct encoding MCA-II-a or MCA-II-d fused to a green fluorescent protein (GFP) under the control of their respective native promoters' The MCA-II-d::GFP results are not shown?
4. Line 133 'We detected GFP fluorescence associated with the ER in embryonic root cells;' This confocal co-localization is plausible but not very convincing. Could the authors present a co-localisation or a similar image with an ER-marker line, e.g. CNX::GFP /CNX::RFP
5. Line 135. 'These findings suggest that MCA-IIs cleave proteins at the ER to regulate their abundance.' I note that in the absence of a signal peptide, MCA-IIs are located, presumably, on the outside of the ER. In this case, it seemed unlikely that MCA-IIs cleaved proteins inside the ER, which is what the COFRADIC showed.
6. Line 155 'These assays showed that MCA-IIs interact weakly with two chaperones involved in proteasome assembly' Is it fair to generalise to all IIs, rather than making a statement for the two tested only? Besides, the next sentence is unclear: I am unsure what 7 and 1 peptides refer to 'regulatory sub 13a, 7 and 1 peptides for MCA-II-b, respectively'; please rewrite.
7. Line 211 'MCA-II-a localised on lipid droplets together with CDC48 (Fig. 3B-E'. need corrected as 3E is about PUX10?
8. Fig 4 A, I don't find the co-localisation of MCA-II-a and CDC48 convincing; the puncta with co-localisation are not really visible in the picture, even blown up on a computer screen. Both fusion proteins seem to mainly not co-localise. Are there better images that could be provided?
9. Fig 4, Please note that the scale bars are the same size in the inset compared to the main photograph, but they should be longer as the inset is a close-up. Or the new value of the scale bar should be indicated.
10. Line 265 'Next, we measured seed viability using tetrazolium staining and found increased PCD in mca-II-KOc'. Tetrazolium is not a marker of PCD. Because the authors are only measuring death

using tetrazolium, it could be programmed or not be programmed. I would suggest that the authors should write cell death rather than PCD in line 265 and then, in the discussion, propose it is ER-stress-induced PCD.

11. Line 265 In addition, could the tetrazolium test be done on fresh mutant seeds as a control to ensure the tetrazolium gets into the mutant seed/embryo? This is to make sure the loss of viability is occurring in dry seeds and not upon water uptake at germination.

12. Discussion: Loss of viability by ER-stress-induced PCD is an elegant hypothesis; however, the dry seed metabolism is mainly inactive with no protein synthesis. Do the authors expect much ERAD / LDAD during the storage of dry seeds? Considering the quiescence of dry seeds, ER-stress-induced PCD during storage might not be the cause of the loss of viability. The control with tetrazolium stain on fresh mutant seeds is important for this. If, after three months, the mutant dry seeds are dead, then the hypothesis of ER-stress-induced PCD needs more explanations how. Could the authors comment on this in their text or revise the hypothesis?

Supp figures

13. Fig S5 G. Why the different sizes of PAG1 between 7-DAG and seeds? Is only one or both correct? Please explain with an arrow.

14. Fig S7 Red arrow in the legend but no red arrow in the Fig. 7A

15. S7C: FY088, the complemented line has low staining in the photographs but a similar intensity to the WT in the relative intensity graph. Could the authors explain that?

16. S8F, which antibody is used with the western on the right, presumably GFP? Does the red star indicate the activation of MC2a-GFP?

End of comments

Reviewer #3 (Remarks to the Author):

The manuscript by Liu et al. claims a discovery of a novel regulatory pathway by which the Arabidopsis type II metacaspases (MCA-II) controls the localization dynamic of CDC48 (ER vs. lipid droplet or LD) by cleaving PUX10, which was recently implicated in the plant (LD)-associated degradation (LDAD) by recruiting CDC48-type AAA-ATPase essential to extract lipid droplet proteins for proteasome degradation. The PUX10-mediated recruitment of MCA-II would lead to cleavage of PUX10, thus inhibiting its ability to recruit CDC40 to LD. As a result, more CDC48 are associated

with ER to enhance ERAD. Therefore, simultaneous elimination of all 6 members of the MCA-II-family in the *mca-II-KO* mutant would prevent PUX10 cleavage, thus enhancing LDAD and reducing ERAD.

This is a quite interesting hypothesis and a manuscript that present strong evidence (biochemistry coupled with genetic/transgenic experiments) for supporting this hypothesis is certainly appropriate for Nature Communications. Unfortunately, the current manuscript is just too preliminary.

I don't think that the data presented in this entire manuscript supported the authors' claim that "Metacaspases participate in the ER stress response". The evidence presented in Fig. 2 for the ER localization of MCA-II is rather weak, especially when analyzing other images of MCA-II-a-fusion proteins in Fig. 3, 4, and Fig. S8. A better (clearly focused with higher-resolution) confocal images with known ER marker (ER-localized RFP or stained with ER-Tracker red dye) are needed to clearly demonstrate the ER-localization of the GFP-tagged MCA-II-a. In addition, a better sucrose gradient ultracentrifugation protocol in the presence or absence of Mg^{2+} , known to be required for the association of ribosomes with the ER, is needed to convincingly show that MCA-II-a is localized at the ER. Given the availability of a quite specific anti-MCA-II-a antibody (shown in Fig. 1B), the authors should use this antibody to examine the endogenous MCA-II-a's potential ER localization. Because all 6 members of the Arabidopsis MCA-II subfamily lacks any detectable signal peptide, transmembrane domain, or other known organelle-localization motifs, it is important that the authors perform a detailed biochemical analysis of its potential mechanism(s) for its hypothesized ER localization (partial membrane insertion, interactions with membrane lipids or integral membrane proteins, inside the ER lumen especially when the authors attempted experiments to test its direct cleavage of the ER luminal proteins such as BIPs and PDIs). If the authors truly believe that *mca-II-KO* mutant is somehow defective in ERAD, they should do a couple of biochemical experiments to prove this important point!

I really don't understand the logics of including lines of 150-226. I am just curious whether the authors have identified oleosins (*AtOLE1-8* or other LDAD substrates) and PUX10 in their proteomic experiments. The experiments truly related to the main conclusion of the manuscript were described in the lines of 227-255. The authors mainly relied on microscopic pictures to present their evidence. Recent studies have shown that the LD-localized PUX10 is important to recruit CDC48 to support LDAD that degrades several LD proteins including *OLE1*. The authors thus need to provide strong biochemical and genetic/transgenic evidence for active MCA-IIs but not their catalytically inactive variants to cleave PUX10 in Arabidopsis plants.

The authors need to provide experimental support for the mechanistic link between enhanced ER recruitment of CDC48 and enhanced ERAD with seed longevity.

Additional concerns:

- 1) The Arabidopsis pux10 mutant should be used for genetic (genetic interaction with the mca-II-KOc), biochemical (enhanced ERAD), and physiology (seed longevity phenotype) experiments.
- 2) The mca-II-KOc mutant is not really a null mutant based on the T-DNA insertion site of the mca-II-d mutant and the RT-PCR data presented in Fig. S1C revealing low level of MCA-II-d transcript.
- 3) The authors cited “Liu, J. X., Srivastava, R., Che, P. & Howell, S. H. An endoplasmic reticulum stress response in Arabidopsis is mediated by proteolytic processing and nuclear relocation of a membrane-associated transcription factor, bZIP28. Plant Cell 19, 4111-4119” paper for their statement that “In plants, ERAD promotes the formation of an impermeable cuticle, a wax layer that attenuates desiccation stress by retaining in the plant body”. As far as I know, there is no experimental data that demonstrated involvement of ERAD in the cuticle formation although there are several publications implicated a role of an Arabidopsis homolog of the yeast Doa10, an ER membrane anchored E3 ligase known to be involved in the yeast ERAD.
- 4) The mac-II-KOc exhibited a weak seed-dormancy phenotype but a seed longevity defect (only a few seeds germinated after 9-month storage at 4oC) that could be complemented by with a proPRS5a::MCA-II-a-mNeon fusion transgene. I am very curious whether the proMCA-II-a: MCAII-a-GFP transgene, which was used to show the ER localization of MCA-II-a, was able to complement the mca-II-KOc mutant.
- 5) The S1 dataset should include the annotation of the identified Arabidopsis proteins.
- 6) I find a bit strange that the authors perform a proteomic study with dry seeds of wild-type and the mac-II-KOc mutant (stored at 4oC for 3 months known to exhibit no seed germination defect as shown in Fig.1E). The authors mentioned in the text that the 3 month-storage was “before mca-II-KOc seeds lose substantial viability”. I wondered why the authors did not perform a time course (3 month, 4 month, 5 month....) to carefully monitor the longevity phenotype that can help to select several different time points for the proteomic experiment.

RESPONSE TO REVIEWERS' COMMENTS

We thank the three reviewers for their comments. A point-by-point response is provided below:

Reviewer #1 (Remarks to the Author):

The manuscript is a beyond reproach and makes great contributions to the understanding of the roles of MCA-II genes. The unravelling of the relationships between these genes and other components of PCD systems and protein synthesis/degradation in seeds is a trove.

Thanks

1.As a reviewer I made a few "ridiculous" corrections in the .pdf file.

We have considered all the suggested corrections. Thanks.

2.As a researcher, I was curious to know if the mature seeds of Arabidopsis used in certain assays (such as protease activity assays) were peeled. This curiosity is born from recent reports of PCD events in seed coats of soybeans and Vigna beans.

We have used mainly intact mature (after-ripened) seeds with fully developed seed coats (testa) except for the results presented in Fig. 4d; this dataset refutes the hypothesis of a seedcoat-imposed dormancy in *mca-II-KO*¹. The seed coat, as well as other extraembryonic tissues (maternal origin), can also affect germination and seedling establishment. Cells of the developing cowpea seed coats undergo PCD, where caspase-like and metacaspase enzymes are active, while during germination additional cells (e.g., aleurone) may undergo PCD². However, in Arabidopsis, no aleurone PCD has been described during germination, though vacuolization has been observed³. Furthermore, the endosperm expands rapidly after fertilization but later it is mechanically ruptured by the expanding embryo, surrounded by a single remaining aleurone-like endosperm layer⁴. If endosperm is not properly disintegrated, embryos have aberrant phenotypes, which were not observed in our work. Finally, the permeability assays were done in fresh seeds using tetrazolium red speaks against major differences in seed coat PCD in mature seeds. We further present an image from SEM on the structural features of the seed coat, and as can be seen, there are no significant differences (Appendix Fig. 1). We succinctly discuss the necessity of further studying other modes of PCD (Lines 145-147; "The involvement of MCA-IIs in more restricted, specific, and tightly controlled cell death programs (e.g., in the seed coat, endosperm, or embryo suspensor17) requires further investigation.").

3.As the authors evaluated the involvement of MCD-II genes on developmental PCD processes only in root cap systems, I wondered if this would be so also in the case of seed coats. But that's just

We chose to analyze the root cap and not the seed coat, because of a lack of robust markers for PCD in the seed coat, so far. We agree that this is an excellent research line encompassing a more detailed analysis of PCD events, as previously done by a co-author of this work⁴.

4.Other than that, I congratulate the authors on such a refined piece of work.

Thanks

Reviewer #2 (Remarks to the Author):

In plants, there is still a lot to learn about factors regulating seed germination and seed longevity. Seed longevity in storage is an important trait for crop seed production and for seed banks. Here the authors make an important contribution to this research area. The role of metacaspases in the above processes has been totally overlooked because of a problem of genetic redundancy that could not be resolved without the availability of metacaspase multi-KO lines. The authors are the first to have dedicated time to creating multi-KO lines for all the MC II genes, and this unlocked a raft of new observations. This is a well-written Ms, with experiments beautifully carried out and displayed. The Ms is an important study that combines and integrates many different approaches. The work brings a key cellular insight into seed longevity. In addition, the data sets generated give several new leads for future studies, which will establish a better understanding of key cellular processes which integrate metacaspases. The new cellular regulation involving a protease that is identified here will be of interest to all cell biologists. There are only a few small issues to resolve, which are listed below. And possibly, the hypothesis to explain the loss of seed viability in the mutant background needs to be refined (point 12).

Thanks, we did our best to resolve the issues raised by the reviewer.

General comment:

1.Why is the MC numbering convention different from the one in TAIR? For example, here, AtMC4 (At1g79340) is AtMC-II-a, and in TAIR, it is AtMC-II-d. This brings confusion when looking at the literature.

As far as I can tell, Ref 5 only indicates the format of the metacaspase names without specifics. Should the authors follow the TAIR annotation?

We now have added (Lines 95-96); we refer the reviewer to our recent work on the nomenclature of metacaspases⁵, and provide a table with the corresponding TAIR accessions (Supplementary Table 1); we refer the reviewer also to the “Data availability” statement (Line 817, “Gene sequence information of MCAs is provided in Supplementary Table 1.”).

2. AtMCA-II-a is systematically tested in various experiments, but there is no rationale to explain why various other metacaspases are tested simultaneously: sometimes b or sometimes d. Could this be explained?

Other MCA-IIs showed similar localizations (presented in revised Fig. 2); due to expression levels in the seed (Geneinvestigator and DIA proteomics, revised Supplementary Fig. 3 and Data 1), we chose MCA-II-a as a representative MCA-II protein (described in Lines 175-190).

Specific comments:

1. Abstract: wrong use of the word dormancy in this context. See Bewley, J.D., 1997. Seed germination and dormancy. The plant cell, 9(7), p.1055. I would think it should be quiescence, not dormancy.

As per the reviewer’s suggestion, we now use the term *quiescence* instead of *dormancy* (as per the suggested relevant ref.⁶).

2. Figure 2 line 665. ‘the proteasomal subunits (GO, protein folding) such as PAG1 (20S proteasome alpha subunit G- 1: 1.3-fold) and PBA1 (20S proteasome subunit beta 1: not found in WT), were verified using α -PAG1 and α -PBA1 antibodies in Fig S5G’. Looking at Fig S5G, it is PAG1 which is absent of WT, present in the mutant, not PBA1, which has the same pattern in WT and mutants.

We now provide more accurate data on this end (revised Supplementary Fig. 10) and have rephrased this part as follows: “We also observed accumulation of some regulatory subunits of the 26S proteasome in *mca-II-KOc*, PAG1 (20S proteasome alpha subunit G- 1: 1.3-fold), and PBA1 (20S proteasome subunit beta 1, increased only after germination). This biochemical phenotype is reminiscent of proteasomal receptor mutants, e.g., mutants of the REGULATORY PARTICLE NON-ATPASE 10 (*rpn10*; Supplementary Fig. 10e-g; ref. 26).” (Lines: 239-244). Please note, that there are two isoforms of PAG1 (Supplementary Fig. 10g legend).

3. Line 130 ‘To determine the localisations of MCA-II-a and MCA-II-d, we generated transgenic plants carrying a translational fusion construct encoding MCA-II-a or MCA-II-d fused to a green fluorescent protein (GFP) under the control of their respective native promoters’. The MCA-II-d::GFP results are not shown?

We apologize for the typo here; we have now corrected the information about MCA-II-a and MCA-II-b (lines 175-181; corresponding Fig. 2). “To test this speculation, we selected MCA-II-a, -b, and -f as representative MCA-II proteins. We generated stable transgenic lines with translational fusions of MCA-II-a, MCA-II-b, or MCA-II-f to green fluorescent protein (GFP) under the control of their respective native promoters (i.e., sequences 2 kb upstream of the start codon). We observed a partial colocalization of GFP fluorescence signal with most of the ER sheets and associated structures (tagged with 35Spro:HDEL-CFP for MCA-II-a and MCA-II-f) in embryonic root cells (Fig. 2b).”.

4. Line 133 ‘We detected GFP fluorescence associated with the ER in embryonic root cells;’. This confocal co-localization is plausible but not very convincing. Could the authors present a co-localisation or a similar image with an ER-marker line, e.g. CNX::GFP /CNX::RFP

We now include results showing colocalization from a line co-expressing MCA-II-a/f with CFP-KDEL (revised Fig. 2b). We refer the reviewer to comment 3, as well.

5. Line 135. ‘These findings suggest that MCA-IIs cleave proteins at the ER to regulate their abundance.’ I note that in the absence of a signal peptide, MCA-IIs are located, presumably, on the outside of the ER. In this case, it seemed unlikely that MCA-IIs cleaved proteins inside the ER, which is what the COFRADIC showed.

We agree and to avoid confusion, we rephrased this part: “As MCA-II-a is likely excluded from the ER lumen, we aimed to further test whether ER luminal proteins are left intact. We thus performed protein degradome analysis of the seed proteome via combined fractional diagonal chromatography (COFRADIC)²³. This approach enables the identification of proteolytic substrates, the corresponding cleavage sites, and the resulting novel N termini (neo-N termini) formed in vivo (Supplementary Fig. 6a, b). We confirmed lysine (K) as the preferable residue precluding the enriched neo-N-termini consistent with the cleavage specificity of MCA-IIs. We though failed to detect any of the enriched ER proteins identified in the *mca-II-KOc* seeds as direct targets of MCA-IIs by a comparative COFRADIC analysis with the WT seeds (Supplementary Data 3). We further confirmed that MCA-II-a localized on the cytoplasmic face of the ER but not in the lumen, using a split-GFP assay in the transient system of *Nicotiana benthamiana* (Supplementary Fig. 7). This assay is based on GFP auto-assembly, when two polypeptides – the “detector” GFP1-10 (residues 1-214; fused to MCA-II-a) and the “tag” GFP11 (residues 215-230; fused to HDEL) – both non-fluorescing on their own, associate spontaneously and produce fluorescence²⁴. Hence, it is highly unlikely that under the conditions

tested, MCA-II-a and presumably other MCA-IIs enter the ER lumen. These findings show that MCA-IIs associate at least transiently with ER-related structures but do not directly access luminal proteins.”. Lines 191-207. Indeed, MCA-IIs are associated with the cytoplasmic face of the ER (e.g., revised Fig. 2 and Supplementary Figs 5 and 18). We also refer the reviewer to the relevant response to reviewer’s #3 comment 3, where we have experimentally addressed the topology of MCA-II-a. We should also note our results do not exclude the possibility of direct implications of MCA-IIs in the lumen, as has been suggested for other ERAD-related proteins⁷. Our data do not allow us to discount the possibility that other MCA-IIs may localize in the lumen, at least, under certain conditions.

6. Line 155 ‘These assays showed that MCA-IIs interact weakly with two chaperones involved in proteasome assembly’ Is it fair to generalise to all IIs, rather than making a statement for the two tested only? Besides, the next sentence is unclear: I am unsure what 7 and 1 peptides refer to ‘regulatory sub 13a, 7 and 1 peptides for MCA-II-b, respectively’; please rewrite.

We agree that generalization is far-fetched here; we have rephrased accordingly: “*These assays showed that MCA-II-a/b co-precipitated with two chaperones involved in proteasome assembly (Supplementary Fig. 10a and Data 6; 2 out of 32 proteins in the proteasome assembly;*”. To avoid confusion, the reference to the semi-quantitation of peptides was removed.

7. Line 211 ‘MCA-II-a localised on lipid droplets together with CDC48 (Fig. 3B-E’. need corrected as 3E is about PUX10?

Thanks, now Fixed. To avoid confusion, the two figures have been split (now the manuscript includes 5 main figures; Fig. 3 and Fig. 4, for CDC48 and PUX10, respectively).

8. Fig 4 A, I don’t find the co-localisation of MCA-II-a and CDC48 convincing; the puncta with co-localisation are not really visible in the picture, even blown up on a computer screen. Both fusion proteins seem to mainly not co-localise. Are there better images that could be provided?

We have now provided more images, assays, and corresponding quantifications (revised Fig. 3); we established colocalization using antibodies against native MCA-II-a and CDC48. The interaction (or association) with CDC48 of MCA-II-a is transient and that is the reason we used the relevant PLA assays which interlock the interactions. We assume that MCA-II-a/CDC48 are part of a non-stoichiometric complex of a very transient nature. Indeed, MCA-I was recently reported to localize in such transient complexes (“condensates”; ref.⁸). This assumption is supported by the increased association of the protease-dead mutants (referred to in the relevant literature as “protease traps”; e.g., ref.⁹) with proteasomal subunits and ubiquitinated proteins. We refer the reviewer to lines 403-412 in the discussion “*MCA-IIs could potentially function in alternative proteostatic pathways and transiently into the ERAD. We should note the transient nature of MCA-IIs associations with the identified proteins of LDAD studied here (i.e., PUX10, CDC48). This finding fits well in a model whereby small non-stoichiometric complexes form, such as phase-separating condensates. Interestingly, MCAs were recently found to localize to stress-induced condensates⁴², and PUXs have been predicted to form condensates⁴³. It is thus tempting to speculate that PUXs/MCAs are part of condensates in seeds. Noteworthy, even very transient interactions between MCAs and their substrates seem sufficient to fulfill proteolytic cleavage events. As was shown in another study, even though MCA-IIs do not associate firmly with the tonoplast, they can still cleave docked peptides there (i.e., PROPEP1)¹⁰.*”.

9. Fig 4, Please note that the scale bars are the same size in the inset compared to the main photograph, but they should be longer as the inset is a close-up. Or the new value of the scale bar should be indicated.

We thank the reviewer for picking this point up (revised Fig. 2b and legend).

10. Line 265 ‘Next, we measured seed viability using tetrazolium staining and found increased PCD in mca-II-KOc’. Tetrazolium is not a marker of PCD. Because the authors are only measuring death using tetrazolium, it could be programmed or not be programmed. I would suggest that the authors should write cell death rather than PCD in line 265 and then, in the discussion, propose it is ER-stress-induced PCD.

We thank the reviewer for picking this point up; the text is now amended “*We thus measured seed viability using tetrazolium staining and found increased cell death in normally aged or exposed to the short-term controlled deterioration treatment (CDT) mca-II-KOc seeds...*” (Lines 373-375).

11. Line 265 In addition, could the tetrazolium test be done on fresh mutant seeds as a control to ensure the tetrazolium gets into the mutant seed/embryo? This is to make sure the loss of viability is occurring in dry seeds and not upon water uptake at germination.

We agree and have performed the corresponding experiment showing that WT and mca-II-KOc seeds show almost the same levels of tetrazolium staining when they are fresh (revised Supplementary Fig. 21a and Fig. 5). We refer the reviewer also to the response to the reviewer’s #1 comment 1, regarding testa cell death.

12. Discussion: Loss of viability by ER-stress-induced PCD is an elegant hypothesis; however, the dry seed metabolism is mainly inactive with no protein synthesis. Do the authors expect much ERAD / LDAD during the storage of dry seeds? Considering the quiescence of dry seeds, ER-stress-induced PCD during storage might not be the cause of the loss of viability. The control with tetrazolium stain on fresh mutant seeds is

important for this. If, after three months, the mutant dry seeds are dead, then the hypothesis of ER-stress-induced PCD needs more explanations how. Could the authors comment on this in their text or revise the hypothesis?

As the metabolism is indeed very slow (but active) in dry seeds (for example, ref. ¹⁰), the cell death program that we have discovered takes months (>3 months; some variability was observed among different labs involved in this paper). Seeds lose dormancy during a period of dry storage called dry after-ripening ^{11, 12}. After-ripening includes active transcription and has major implications for crop domestication and seed behavior during dormancy cycling in natural ecosystems. We suspect that the MCA-II mutants fail to retain dormancy, and rather enter this state. This partial lack of dormancy could explain the premature germination of fresh seeds (revised Fig. 1d). Accordingly, seeds exit dormancy and proceed to after-ripening as a last chance for germination before accumulating excessive damage and die ¹³. In addition, we tested seed viability by subjecting seeds to an accelerated aging protocol (called “controlled deterioration treatment, CDT”; revised Fig. 5 and Supplementary Fig. 21a-e; “*We thus measured seed viability using tetrazolium staining and found increased cell death in normally aged or exposed to the short-term controlled deterioration treatment (CDT) mca-II-KOc seeds (Fig. 5g). We could partially complement this phenotype using the constructs RPS5a:MCA-II-a-GFP, 35Spro:MCA-II-a-GFP or MCA-II-a:MCA-II-a-GFP, but not the corresponding inactive variant (MCA-II-aPD-GFP; Supplementary Fig. 21a-d). On the contrary, as expected, the lack of PUX10 did not show compromised seed-longevity or sensitivity to CDT (Supplementary Fig. 21e).*”). In this protocol, seeds are incubated at high temperatures and high relative humidity¹³; we confirmed the results observed with prolonged storage of Fig. 1. Finally, we discount the possibility that the observed phenotypes are due to compromised seed filling that could compromise the energy reserves of seeds (and has been shown as a potential function of the ERAD modulating protein UBC32; ref. ¹⁴), as revealed using experiments of N¹⁵ feeding (Appendix Fig. 2). Interestingly, ERAD components accumulate in seeds (such as the luminal ERAD components PAWH1/2 ¹⁵; see Supplementary Data 1 and 4 from DIA analyses confirming the expression of these proteins in seeds). As a cautionary note, we do not believe that PCD is solely induced by ER stress (i.e., *ire1a/b* mutants do not show exaggerated CDT; revised Supplementary fig. 9), but rather by a far more complex scenario, which likely involves the fatty acids that reduce longevity in combination with attenuated proteostasis (e.g., faulty ERAD activity), oil droplets-fatty acids that exert mechanical stress on membranes and likely more general membrane homeostasis executed by MCA-IIs (discussion lines 432-445; “*Furthermore, lipid droplets in animal cells link to an upregulated ERAD through fatty acids, providing a mechanistic explanation for the Mediterranean diet-dependent lifespan extension in humans*⁴⁵. *In tocopherol-deficient Arabidopsis mutants, oxidized fatty acids accumulate showing decreased seed longevity*⁴⁶. *It is possible that like *vte2*, alterations in the ERAD/LDAD dynamics cause lipid oxidation which might contribute to the seed longevity phenotype of *mca-II-KOc* mutants. Spatial control of lipid droplet proteins with similar features to oleosins has been reported in yeast for a *DOA10* homolog⁴⁷. Hence, the investigation of further links between MCAs and the cytoplasmic ERAD pathway (e.g., *DOA10*) is justified, also including the type I MCAs. Still, the pathway linking MCA-II-dependent PUX10 cleavage and CDC48 activity appears to have evolved as an elegant mechanism enabling sustained protein production while maintaining protein homeostasis, - a crucial prerequisite for seed development.*”). We further address this point genetically, using the single *pux10* and septuple *pux10 mca-II-KOc* mutants (revised Fig. 5 and Supplementary Fig. 21). Remarkably, lipid droplets show extended surface tension with rigidity that can perturb and fenestrate membranes ¹⁶.

Supp figures

13. Fig S5 G. Why the different sizes of PAG1 between 7-DAG and seeds? Is only one or both correct? Please explain with an arrow.

Thanks for picking this point up; two gene models are leading to the expression of 2 different isoforms with the sizes we determined (from gene models AT2G27020.1 and 2 with predicted protein sizes, 27.3 kDa and 39.3 kDa, respectively). We provide this information in the corresponding figure legend (Supplementary Fig. 10g). We would not like to discuss further this point in the manuscript, as it is too early to suggest potential reasons for this differential expression.

14. Fig S7 Red arrow in the legend but no red arrow in the Fig. 7A

Please see the next comment (data removed and appended in Fig. 3 herein).

15. S7C: FY088, the complemented line has low staining in the photographs but a similar intensity to the WT in the relative intensity graph. Could the authors explain that?

We append below the way the calculations were conducted (Appendix Fig. 3). Please, note that this dataset, according to the response to reviewer #3 point 3 in “additional concerns” has now been removed. The role of DOA10 in the establishment of non-permeable cuticles is rather controversial, as the lack of water loss in this mutant relates to stomatal ledges inhibiting evaporation. One would expect the “oilier” cuticle of the *doa10* mutant, albeit thicker, to be more permeable ¹⁷. Furthermore, recent data show that DOA10 is not functionally

equivalent to its yeast homolog and lost the ability to work in the proteasome-linked N-end rule pathway¹⁷. The links we have provided with DOA10 in Arabidopsis merit further exploration (revised Supplementary Fig. 18); this is justified further considering the role of DOA10 in yeast in lipid droplets (e.g., ref.¹⁸). Furthermore, the lack of *hrd1a/b* leads to a less waxy cutin, a slightly contradicting phenotype compared to that of the DOA10 mutant (*cer9*)¹⁹. In sum, the links between ERAD and cuticle, are not clear enough.

16. S8F, which antibody is used with the western on the right, presumably GFP? Does the red star indicate the activation of MC2a-GFP?

The figure was amended accordingly (now revised Supplementary fig. 17a). Thanks for picking this point up. The red asterisk indicates autocleavage.

Reviewer #3 (Remarks to the Author):

The manuscript by Liu et al. claims a discovery of a novel regulatory pathway by which the Arabidopsis type II metacaspases (MCA-II) controls the localization dynamic of CDC48 (ER vs. lipid droplet or LD) by cleaving PUX10, which was recently implicated in the plant (LD)-associated degradation (LDAD) by recruiting CDC48-type AAA-ATPase essential to extract lipid droplet proteins for proteasome degradation. The PUX10-mediated recruitment of MCA-II would lead to cleavage of PUX10, thus inhibiting its ability to recruit CDC40 to LD. As a result, more CDC48 are associated with ER to enhance ERAD. Therefore, simultaneous elimination of all 6 members of the MCA-II-family in the *mca-II-KOc* mutant would prevent PUX10 cleavage, thus enhancing LDAD and reducing ERAD. This is a quite interesting hypothesis and a manuscript that present strong evidence (biochemistry coupled with genetic/transgenic experiments) for supporting this hypothesis is certainly appropriate for Nature Communications.

1. Unfortunately, the current manuscript is just too preliminary. I don't think that the data presented in this entire manuscript supported the authors' claim that "Metacaspases participate in the ER stress response".

We thank the reviewer for the constructive criticism; we did our best to address all points. Regarding the title of the session, we agree that it was not accurate enough [now reads: "*Type II metacaspases participate in a UPR-independent proteostasis pathway*" (line 164)].

2. The evidence presented in Fig. 2 for the ER localization of MCA-II is rather weak, especially when analyzing other images of MCA-II-a-fusion proteins in Fig. 3, 4, and (Supplementary Fig. 8. A better (clearly focused with higher-resolution) confocal images with known ER marker (ER-localized RFP or stained with ER-Tracker red dye are needed to clearly demonstrate the ER-localization of the GFP-tagged MCA-II-a. In addition, a better sucrose gradient ultracentrifugation protocol in the presence of absence of Mg²⁺, known to be required for the association of ribosomes with the ER, is needed to convincingly show that MCA-II-a is localized at the ER. Given the availability of a quite specific anti-MCA-II-a antibody (shown in Fig. 1b), the authors should use this antibody to examine the endogenous MCA-II-a's potential ER localization.

We improved images and performed further experiments/analyses; we also made necessary textual amendments. In more detail:

1. Additional analyses of MCA-II-a-GFP localization: we noticed that cells differ in their expression levels in MCA-IIs (quantified in revised Supplementary fig. 5), and when MCA-II-a expression was too high, the localization to the ER was barely visible due to high diffusion of GFP signal. We believe that this is due to the saturation of binding sites on the ER surface, which leads to the free diffusion of MCA-II-a molecules; other models for regulation of MCA-IIs localization are also possible (e.g., Ca²⁺-dependent regulation of localization; some cells have more than others, and Ca²⁺ has been found to regulate the assembly of proteins at the plasma membrane; ref.²⁰).

2. Examination of the native localization of MCA-II-a: we used the specific anti-MCA-II-a antibody (verified further as shown by the "uncropped blots-relevant to fig. 1 page") for colocalizations with the ER-localized BiP, showing also the association with ER-associated structures (e.g., oil droplets; the association of oil droplets with the ER is discussed extensively in refs.²¹), confirming the live imaging results (revised Fig. 2b and Supplementary Fig. 5; corresponding quantifications).

3. Topology assessment of MCA-II-a with imaging: we tested interactions among MCA-II-a and the ER-specific DOA10 or the luminal PDI5 (Supplementary Fig. 18; albeit part of the protein can be outside the lumen as has been reported in animal cells, while some proteins can show topological inversions; ref.¹⁶ and its supplemental discussion²²). Furthermore, we used split-GFP experiments to interlock transient interactions at the ER and examine whether MCA-II-a can be found in the lumen (Supplementary Fig. 17). Our results show that MCA-II-a is at the cytoplasmic face of the ER where it can interact with the ER-associated protein DOA10a, a minor component of the ERAD¹⁷; this association does not necessarily connote a direct role in this ERAD pathway. As a cautionary note, we cannot discount the possibility that other MCA-IIs (under certain conditions) can localize in the lumen. Further research on the links between individual MCA-IIs (e.g., the secreted MCA-II-d) and the ER/LDAD/ERAD is merited.

4. Biochemistry for the association of MCA-II-a with the ER: we refined and re-performed fractionation analyses using Mg²⁺ (per the ref. ¹⁵). We refer to the reviewer on the new WBs showing the fractionation of the native MCA-II-a (revised Fig. 2c). Furthermore, in Supplementary Fig. 5e, d we have performed solubilization assay from microsomes and protease protection assays. Both assays show a complex interaction of MCA-II-a with the ER (e.g., through electrostatic bonds and interactions with ER-membrane integral proteins).

5. Textual amendments and further discussion: we discuss the possibility of the existence of a “kiss-and-run” mechanism for substrate cleavage by MCA-IIs. This mechanism has also been previously exemplified: even though MCA-IIs do not associate with the tonoplast, they can still cleave docked peptides there (i.e., PROPEP1) ²³. We further highlight that MCA-IIs constitute an auxiliary pathway for the ERAD (and other homeostatic mechanisms) that likely functions when there is a need, for example, in the demanding environment of the seed. The relevant discussion is quoted as “MCAs could potentially function in alternative proteostatic pathways and transiently into the ERAD. We should note the transient nature of MCA-IIs associations with the identified proteins of LDAD studied here (i.e., PUX10, CDC48). This finding fits well in a model whereby small non-stoichiometric complexes form, such as phase-separating condensates. Interestingly, MCAs were recently found to localize to stress-induced condensates⁸, and PUXs have been predicted to form condensates²⁴. It is thus tempting to speculate that PUXs/MCAs are part of condensates in seeds. Noteworthy, even very transient interactions between MCAs and their substrates seem sufficient to fulfill proteolytic cleavage events. As was shown in another study, even though MCA-IIs do not associate firmly with the tonoplast, they can still cleave docked peptides there (i.e., PROPEP1)” and “Finally, we recognize that our study is not exhaustive and parallel pathways could be relevant to the observed phenotypes. For example, we opted for using bri1-9, due to the links of MCA-IIs with Ub-proteins. We should note though that Ub is not always linked to ERAD, as has been shown for other substrates like bri1-5 and the SUBEX-C56Y degradation³⁸. Furthermore, lipid droplets in animal cells link to an upregulated ERAD through fatty acids, providing a mechanistic explanation for the Mediterranean diet-dependent lifespan extension in humans⁴⁵. In tocopherol-deficient Arabidopsis mutants, oxidized fatty acids accumulate showing decreased seed longevity⁴⁶. It is possible that like vte2, alterations in the ERAD/LDAD dynamics cause lipid oxidation which might contribute to the seed longevity phenotype of mca-II-KOc mutants. Spatial control of lipid droplet proteins with similar features to oleosins has been reported in yeast for a DOA10 homolog⁴⁷. Hence, the investigation of further links between MCAs and the cytoplasmic ERAD pathway (e.g., DOA10) is justified, including also the type I MCAs. Still, the pathway linking MCA-II-dependent PUX10 cleavage and CDC48 activity appears to have evolved as an elegant mechanism enabling sustained protein production while maintaining protein homeostasis, - a crucial prerequisite for seed development.”.

3. Because all 6 members of the Arabidopsis MCA-II subfamily lacks any detectable signal peptide, transmembrane domain, or other known organelle-localization motifs, it is important that the authors perform a detailed biochemical analysis of its potential mechanism(s) for its hypothesized ER localization (partial membrane insertion, interactions with membrane lipids or integral membrane proteins, inside the ER lumen especially when the authors attempted experiments to test its direct cleavage of the ER luminal proteins such as BIPs and PDIs).

Despite the exact mechanism of this association being far beyond the scope of our work, we attempted to address this comment:

1. Biochemical experiment: we did an association assay with microsomes, as done for the ERAD components PAWH1/2¹⁵. We observed that detergents or salt promoted the release of MCA-II-a from microsomes, suggesting that the MCA-II-a is likely interacting with an ER-embedded protein, the DOA10 (revised Supplementary Figs 5 and 18). Other, yet unknown components could be involved, and we believe that both DOA10 and MCA-II-a (and PUX10) are parts of a larger complex. We would also like to note that a recent work in *Chlamydomonas*, while this work was under revision, highlighted the association of MCAs with membranes from co-authors of this work ²⁴.

2. Imaging experiments: we exclude the possibility that MCA-II-a is extensively localized in the lumen, as we could not see interaction in split-GFP experiments using the HDEL marker (revised Supplementary fig. 7), and the overall signal of MCA-II-a around ER-sheets showed higher spreading (revised Fig. 2 and Supplementary fig. 18). Yet, we cannot discount the possibility that additional members of the family could localize in the ER lumen. Hence, further work in this direction is required.

4. If the authors truly believe that mca-II-KOc mutant is somehow defective in ERAD, they should do a couple of biochemical experiments to prove this important point!

We agree and we have addressed extensively this point:

1. Drug treatments for testing ERAD-specific activity: we have performed drug treatments with compounds known to sensitize ERAD mutants, namely tunicamycin, thapsigargin, and kifunensin ^{25, 26}. Both *mca-II-KOc* and *pux10* were more sensitive compared to the WT (Supplementary Fig. 21). In addition to increased salt

sensitivity, as observed in ERAD mutants, *mca-IIs* showed increased sensitivity to salt treatments (revised Supplementary Fig. 19) and to an HRD1 inhibitor (synoviolin; Appendix Fig. 4). It has been observed that UBC32 which is not a luminal protein, targets OS9 which is HDR1-associated, a luminal ERAD component²⁷.

2. Genetics for testing the ERAD activity: we selected the ERAD substrate Bri1-9, known to require a functioning ERAD pathway for retention at the ER (revised Fig. 5). We further strengthened the lack of links to the UPR, adding up genetic data (*ire1a/b*; revised Supplementary Fig. 9). “*More recent data suggest a possible clearance of UPR proteins by the ERAD²⁵, suggesting a possible antagonistic relationship between the two pathways, which is also consistent with our observations here (discussed also in Lines 260-263; “Furthermore, ERAD by directly targeting UPR components may downregulate it, for example, in older seeds (Supplementary Data 4).”*”.

3. Textual amendments: We further provide a cautionary note that Ub-moieties are not necessarily linked to ERAD, as has been shown for *bri1-5* and the SUBEX-C56Y degradation (Line 430; “*We should note though that Ub is not always linked to ERAD, as has been shown for other substrates like bri1-5 and the SUBEX-C56Y degradation*”).”.

5. I really don't understand the logics of including lines of 150-226. I am just curious whether the authors have identified oleosins (*AtOLE1-8* or other LDAD substrates) and PUX10 in their proteomic experiments.

We have rewritten this part to improve clarity (now lines 192-252). The message that we wish to convey is that while a canonical UPR is not induced, MCA-IIs still participate in a proteostatic pathway. We show that:

1. UPR is not significantly induced: despite the observed effects in vacuolar morphology and the accumulation of aggregates of *mca-II-KOc*, we could not establish a strong link to the UPR. We discounted the induction of UPR signatures at both RNA and proteome levels (revised Supplementary Data 4).

2. MCA-IIs regulate alternative proteostatic pathways: Some proteasomal subunits increase in the MCA-IIs mutant (revised Supplementary Fig. 10). This result implies a cell compensatory function for an increased ER load²⁸. Furthermore, MCA-IIs can be found in complexes with ubiquitinated proteins. We believe that this association involves likely the cleavage of Ub-proteins through UPS.

3. Links to the proteostatic LDAD pathway: we provide data on the levels of oleosin proteins (and other relevant proteins) from our new proteomic protocol in which we implemented a new seed-specific DIA approach to improve proteome coverage. **This approach allowed us to capture for the first time an unprecedented number of protein hits**, including the peptide of PUX5 identified by COFRADIC, and confirm it in the DIA approach (Appendix Fig. 5). We used a pipeline for quantitation of peptides that allowed us to identify PUX10 as mainly affected by a lack of MCAs. To our knowledge, the only known LDAD-PUX10 substrates are OLE proteins²⁹. We refer to the reviewer of our revised Fig. 3, where OLEs (and other relevant proteins in Supplementary Data 7; ref.³⁰) are shown and quantified.

4. Textual amendments: we discuss the possibility of more complex models, whereby additional links of PUXs with MCA-II exist. Given the results of COFRADIC on the levels of PUX5 peptide, a nuclear protein, and the fact that MCA-II-a also gets into the nucleus, this is further merited. Furthermore, we discuss that PUX10 has a unique structure and resembles UBXD8 (Ubiquitin regulatory X domain-containing protein 8), a mammalian CDC48 adaptor that is similar in architecture to PUX10 but no other PUXs is regulated by a conserved in plants rhomboid pseudoprotease (UBA-domain containing, UBAC2; common features with Derlins), which is responsible for shuttling CDC48 between the ER and lipid droplets, thereby regulating energy squandering^{31,32}. PUX10 is the only known protein of the family with an amphipathic helix and thus with the ability to bind on membranes^{33 34} (relevant lines 400-403).

6. The experiments truly related to the main conclusion of the manuscript were described in the lines of 227-255. The authors mainly relied on microscopic pictures to present their evidence. Recent studies have shown that the LD-localized PUX10 is important to recruit CDC48 to support LDAD that degrades several LD proteins including OLE1. The authors thus need to provide strong biochemical and genetic/transgenic evidence for active MCA-IIs but not their catalytically inactive variants to cleave PUX10 in Arabidopsis plants. We agree and in the revised version we provide:

1. Biochemical experiments: as per the request of the reviewer, we introgressed PUX10-GFP under native promoter in *mca-II-KOc*; western blot showed PUX10 cleavage mainly in WT and reduced stability (revised Fig. 4a-c), confirming the data from proteomics (Supplementary Data 1 and 9). The mutant MCA-II variant did not cleave PUX10 (revised Fig. 4c). Unfortunately, an antibody produced against PUX10 was not functional in our hands (Appendix Fig. 6).

2. Trangenic/genetic experiments: we show that *pux10 mca-II-KOc* is partially rescued compared to the independent mutants (revised Supplementary Figs. 17 and 21); OLE1 seems to be the preferable substrate for PUX10, although why this is happening needs further experimental verification in future works.

7. The authors need to provide experimental support for the mechanistic link between enhanced ER recruitment of CDC48 and enhanced ERAD with seed longevity.

We agree and have provided further support. We present evidence that *pux10* mutant, although shows slight sensitivity to ERAD drugs, is not affected by CDT like *mca-II-KOc*. We provide evidence that both ERAD/oil droplet dynamics affect seed longevity (Supplementary Fig. 21). **pux10 seems epistatic to the mca-II-KOc mutant, as it can partially rescue the ERAD, oleosin levels and seed longevity phenotype.** Furthermore, to clarify, the *pux10* mutant does not show enhanced ERAD albeit showing similar to WT CDC48 recruitment at the ER; enhanced ERAD as in mutant *ubc32* did not affect the longevity of seeds in a CDT setting (Appendix Fig. 7). Please also note that although the phenotype of *mca-II-KOc pux10* is rescued in the CDT, we did not find a similar rescue of general proteostasis (i.e., tunicamycin sensitivity; Supplementary Fig. 19). These results further confirm that UPR is not the main pathway for seed longevity.

Additional concerns:

1) The Arabidopsis *pux10* mutant should be used for genetic (genetic interaction with the *mca-II-KOc*), biochemical (enhanced ERAD), and physiology (seed longevity phenotype) experiments.

We refer the reviewer to points 5 and 6 above.

2) The *mca-II-KOc* mutant is not really a null mutant based on the T-DNA insertion site of the *mca-II-d* mutant and the RT-PCR data presented in Fig. S1C revealing low level of MCA-II-d transcript.

We apologize for this confusion. We now provide the exact way that the mutant was built (revised Supplementary Fig. 1 and revised Fig. 1). We discuss this further in lines 97-111: “To overcome the potential redundancies among MCA-IIs, we used clustered regularly interspaced short palindromic repeat (CRISPR)/CRISPR-associated nuclease 9 (Cas9)-mediated genome editing. First, we introduced four mutations in *mca-II-a/b/c/e* using the T-DNA double mutant *mca-II-df* as background (Supplementary Fig. 1a-c). We found that the *mca-II-d* mutant allele carried a T-DNA insertion upstream of the first start codon and showed reduced but detectable levels of *mca-II-d* mRNA (Supplementary Fig. 1d). To remove residual *mca-II-d* levels, apart from *mca-II-a/b/c/e*, we sequentially targeted *mca-II-d* through CRISPR in the *mca-II-a/b/c/e/f* mutant background. After screening \square 1,000 individual plants, we obtained plants harboring single or higher-order homozygous mutations in the MCA-II genes in various combinations, as well as two independent lines with homozygous mutations in all six MCA-IIs. These sextuple MCA-II mutants were crossed to wild type (WT) to remove the Cas9 transgenes, and all mutations were validated in the F2 generation; we refer to these mutants hereafter as *mca-II-KOc* (“c” for “clean line” without Cas9 transgenes; Supplementary Fig. 1e).”

3) The authors cited “Liu, J. X., Srivastava, R., Che, P. & Howell, S. H. An endoplasmic reticulum stress response in Arabidopsis is mediated by proteolytic processing and nuclear relocation of a membrane-associated transcription factor, bZIP28. *Plant Cell* 19, 4111-4119” paper for their statement that “In plants, ERAD promotes the formation of an impermeable cuticle, a wax layer that attenuates desiccation stress by retaining in the plant body”. As far as I know, there is no experimental data that demonstrated involvement of ERAD in the cuticle formation although there are several publications implicated a role of an Arabidopsis homolog of the yeast Doa10, an ER membrane anchored E3 ligase known to be involved in the yeast ERAD. We thank the reviewer for noticing. We have removed this part, as the cuticle development and the associated phenotypes are rather controversial given that DOA10a (*SUD1/CER9/DOA10A*) link to enhanced drought tolerance is likely associated with deposits of cuticle on the stomatal aperture (cuticular ledges) that reduce water evaporation³⁵, rather than a direct effect of the changes in cuticle properties. Furthermore, the shift in the wax composition of the cuticle towards fatty acids in *doa10* mutants would likely increase the permeability of the barrier³⁵. The formation of the cuticle is a very complex phenomenon that involves an interplay between embryonic and extraembryonic tissues³⁶, and merits further investigation. We refer the reviewer to the response provided to reviewer 2, points 14 and 15.

4) The *mca-II-KOc* exhibited a weak seed-dormancy phenotype but a seed longevity defect (only a few seeds germinated after 9-month storage at 4oC) that could be complemented by with a proPRS5a::MCA-II-a-mNeon fusion transgene. I am very curious whether the proMCA-II-a: MCAII-a-GFP transgene, which was used to show the ER localization of MCA-II-a, was able to complement the *mca-II-KOc* mutant.

As per the reviewer's request, we performed this experiment (revised Supplementary Fig. 21c). We show that the native promoter-driven MCA-II-a could also rescue germination phenotype, similarly to RPS5apro.

5) The S1 dataset should include the annotation of the identified Arabidopsis proteins.

Done, thanks for raising the issue (revised Supplementary Data 1).

6) I find a bit strange that the authors perform a proteomic study with dry seeds of wild-type and the *mca-II-KOc* mutant (stored at 4oC for 3 months known to exhibit no seed germination defect as shown in Fig. 1E). The authors mentioned in the text that the 3 month-storage was “before *mca-II-KOc* seeds lose substantial viability”. I wondered why the authors did not perform a time course (3 month, 4 month, 5 month....) to

carefully monitor the longevity phenotype that can help to select several different time points for the proteomic experiment.

We agree with this point and have re-designed our experimentation using a DIA approach for proteomics to get more datasets at two-time points (Supplementary Data 1). We would not, at this stage, like to address the situation in more mature seeds, as cell death would likely skew total proteomic profiles and may lead to false interpretations.

APPENDIX

Appendix Fig. 1. Seed coat in *mca-II-KOc* does not show aberrant phenotypes.

SEM images are from a single experiment.

Appendix Fig. 2. Chase of N15-glutamate during seed filling.

We examined whether seed storage profiles in the MCA-II-KOc. Isotopic tracing using heavy nitrogen (N15) coupled with GC-tandem MS analyses, showed that MCA-II showed has a normal nutrient channeling from maternal tissues (known as “seed filling”) to seeds which would otherwise have led to reduced seed storage proteins. P values were calculated by one-way ANOVA (N = 5 biological replicates).

Appendix Fig. 3. Cuticle accumulation in *mca-II-KOoc*.

(a) Sites for quantification of cuticle of FY-088 staining staining for two-day-post-germination seedlings of WT, *mca-II-KOoc*, and a complementation line (Com., rescued with *RPS5apro:MCA-II-a-mNeon*; 9-month-old seeds). The dotted rectangular indicates the site of quantification. (b) Loss of apical hook in *mca-II-KOoc* and reduced hypocotyl length. The etiolated seedlings were grown for 5 days under darkness. Red arrowheads denote the hypocotyl, while yellow arrowheads the apical hook. Right: quantification of the hypocotyl length (RU, relative units). The data are from three experiments, and P-values were calculated by ordinary one-way ANOVA (N = 3, n = 10 hypocotyls). (c) Representative image of seven-days-after-germination seedlings grown on either mock (DMSO), or ABA-containing plates (100 nM; lack of cuticle leads to increased ABA susceptibility). The experiment was repeated three times with similar results (N = 3, n = 8). (d) Representative micrograph of FY-088 staining for two-day-post-germination seedlings of WT, *mca-II-KOoc*, and a complementation line (Com., rescued with *RPS5apro:MCA-II-a-mNeon*; 9-month-old seeds). The data are from three experiments, and P-values were calculated by ordinary one-way ANOVA (N = 3, n = 33 intensity measurements). (e) Representative images from cuticle permeability tests of etiolated seedlings 4-days post germination (DPG) of WT (9-month-old seeds), *mca-II-KOoc* (fresh seeds, 3-, or 9-month-old), and a complementation line (Com., rescued with *RPS5apro:MCA-II-a-mNeon*; 9-month-old seeds) stained with toluidine blue (0.05% [w/v]). 10 Scale bar, 0.5 cm.

Appendix Fig. 4. Synoviolin treatment.

(a) Bri1-9 upon synoviolin (LS-102) treatment gets stabilized and localizes to the plasma membrane. Confocal micrographs from epidermal root cells of an experiment repeated twice with similar results. Scale, 50 μ m. (b) Quantifications of root length for WT, *mca-II-KOc* or *ire1a ire1b* for synoviolin-treated seedlings.

Appendix Fig. 5. Levels of the N-terminal peptide of PUX5 in COFRADIC experiment.

In the COFRADIC data, the N terminus of a PUX homolog (PUX5) was highly enriched in WT but not in *mca-II-KOc* samples (Supplementary Data 2). Yet, DIA showed that quantification of additional peptides did not yield similar results. The peptide used GGAGENKETENPSGIR was also found in WT in DIA but not in *mca-II-KOc*.

Appendix Fig. 6. PUX10 antibody.

Immunoblot analysis using an α -PUX10 in *mca-II-KO*c (line 30) and the hypomorphic *mca-II-KO* (line 27, with residual *mca-II-e*), WT (*Col-0*), and *pux10-1*. A specific band of not expected size is shown. Note in the size ca. 14 kDa, oleosin protein, increased in PUX10 and reduced in *mca-II-KO*c.

Mock

TM 10 μ M

CDT

Appendix Fig. 7. Enhanced ERAD in *ubc32* did not affect CDT.

Tunicamycin was used here to check for the sensitivity of *ubc32* (TM 10 nm). Images are from a single experiment.

REFERENCES

1. Lima NB, *et al.* Programmed cell death during development of cowpea (*Vigna unguiculata* (L.) Walp.) seed coat. *Plant Cell Environ* **38**, 718-728 (2015).
2. Lemos Rocha G, *et al.* Programmed cell death in soybean seed coats. *Plant Sci* **288**, 110232 (2019).
3. Haughn G, Chaudhury A. Genetic analysis of seed coat development in Arabidopsis. *Trends Plant Sci* **10**, 472-477 (2005).
4. Doll NM, *et al.* Endosperm cell death promoted by NAC transcription factors facilitates embryo invasion in Arabidopsis. *Curr Biol* **33**, 3785-3795.e3786 (2023).
5. Minina EA, *et al.* Classification and Nomenclature of Metacaspases and Paracaspases: No More Confusion with Caspases. *Mol Cell* **77**, 927-929 (2020).
6. Bewley JD. Seed Germination and Dormancy. *The Plant Cell* **9**, 1055-1066 (1997).
7. Strasser R. Protein Quality Control in the Endoplasmic Reticulum of Plants. *Annu Rev Plant Biol* **69**, 147-172 (2018).
8. Ruiz-Solaní N, *et al.* Arabidopsis metacaspase MC1 localizes in stress granules, clears protein aggregates, and delays senescence. *Plant Cell* **35**, 3325-3344 (2023).
9. Rei Liao J-Y, van Wijk KJ. Discovery of AAA+ Protease Substrates through Trapping Approaches. *Trends Biochem Sci* **44**, 528-545 (2019).
10. Sallon S, *et al.* Germination, genetics, and growth of an ancient date seed. *Science* **320**, 1464 (2008).
11. Carrera E, *et al.* Seed after-ripening is a discrete developmental pathway associated with specific gene networks in Arabidopsis. *Plant J* **53**, 214-224 (2008).
12. Holdsworth MJ, Bentsink L, Soppe WJJ. Molecular networks regulating Arabidopsis seed maturation, after-ripening, dormancy and germination. *New Phytologist* **179**, 33-54 (2008).
13. Kim W, Zeljković SĆ, Piskurewicz U, Megies C, Tarkowski P, Lopez-Molina L. polyamine uptake transporter 2 (*put2*) and decaying seeds enhance phyA-mediated germination by overcoming PIF1 repression of germination. *PLOS Genetics* **15**, e1008292 (2019).
14. Tang S, *et al.* An E2-E3 pair contributes to seed size control in grain crops. *Nat Commun* **14**, 3091 (2023).
15. Lin L, *et al.* PAWH1 and PAWH2 are plant-specific components of an Arabidopsis endoplasmic reticulum-associated degradation complex. *Nat Commun* **10**, 3492 (2019).
16. Ivanovska IL, Tobin MP, Bai T, Dooling LJ, Discher DE. Small lipid droplets are rigid enough to indent a nucleus, dilute the lamina, and cause rupture. *J Cell Biol* **222**, (2023).
17. Etherington RD, *et al.* Nt-acetylation-independent turnover of SQUALENE EPOXIDASE 1 by Arabidopsis DOA10-like E3 ligases. *Plant Physiol* **193**, 2086-2104 (2023).

18. Ruggiano A, Mora G, Buxó L, Carvalho P. Spatial control of lipid droplet proteins by the ERAD ubiquitin ligase Doa10. *Embo J* **35**, 1644-1655 (2016).
19. Wu P, Gao H, Liu J, Kosma DK, Lü S, Zhao H. Insight into the roles of the ER-associated degradation E3 ubiquitin ligase HRD1 in plant cuticular lipid biosynthesis. *Plant Physiol Bioch* **167**, 358-365 (2021).
20. Diaz M, *et al.* Calcium-dependent oligomerization of CAR proteins at cell membrane modulates ABA signaling. **113**, E396-E405 (2016).
21. Tang S, *et al.* Mechanism-based traps enable protease and hydrolase substrate discovery. *Nature* **602**, 701-707 (2022).
22. Chen Q, Denard B, Lee CE, Han S, Ye JS, Ye J. Inverting the Topology of a Transmembrane Protein by Regulating the Translocation of the First Transmembrane Helix. *Mol Cell* **63**, 567-578 (2016).
23. Hander T, *et al.* Damage on plants activates Ca(2+)-dependent metacaspases for release of immunomodulatory peptides. *Science* **363**, (2019).
24. Zou Y, *et al.* Thermoprotection by a cell membrane-localized metacaspase in a green alga. *The Plant Cell*, (2023).
25. Wang Y, *et al.* Selective degradation of an ER stress-induced protein by ER-associated degradation mechanism during stress recovery. *bioRxiv*, 2022.2006.2006.495033 (2022).
26. Liang F, Sze H. A high-affinity Ca²⁺ pump, ECA1, from the endoplasmic reticulum is inhibited by cyclopiazonic acid but not by thapsigargin. *Plant Physiol* **118**, 817-825 (1998).
27. Chen Q, Liu R, Wang Q, Xie Q. ERAD Tuning of the HRD1 Complex Component AtOS9 Is Modulated by an ER-Bound E2, UBC32. *Mol Plant* **10**, 891-894 (2017).
28. Motosugi R, Murata S. Dynamic Regulation of Proteasome Expression. *Front Mol Biosci* **6**, 30 (2019).
29. Deruyffelaere C, *et al.* PUX10 Is a CDC48A Adaptor Protein That Regulates the Extraction of Ubiquitinated Oleosins from Seed Lipid Droplets in Arabidopsis. *The Plant Cell* **30**, 2116-2136 (2018).
30. Shao Q, Liu X, Su T, Ma C, Wang P. New Insights Into the Role of Seed Oil Body Proteins in Metabolism and Plant Development. *Front Plant Sci* **10**, 1568 (2019).
31. Olzmann JA, Richter CM, Kopito RR. Spatial regulation of UBXD8 and p97/VCP controls ATGL-mediated lipid droplet turnover. *Proc Natl Acad Sci U S A* **110**, 1345-1350 (2013).
32. Wang C-W, Lee S-C. The ubiquitin-like (UBX)-domain-containing protein Ubx2/Ubx8 regulates lipid droplet homeostasis. *J Cell Sci* **125**, 2930-2939 (2012).
33. Kretschmar FK, *et al.* PUX10 Is a Lipid Droplet-Localized Scaffold Protein That Interacts with CELL DIVISION CYCLE48 and Is Involved in the Degradation of Lipid Droplet Proteins. *Plant Cell* **30**, 2137-2160 (2018).

34. Zhang J, Vancea AI, Arold ST. Targeting plant UBX proteins: AI-enhanced lessons from distant cousins. *Trends in Plant Science* **27**, 1099-1108 (2022).
35. Zhao H, *et al.* The Putative E3 Ubiquitin Ligase ECERIFERUM9 Regulates Abscisic Acid Biosynthesis and Response during Seed Germination and Postgermination Growth in Arabidopsis *Plant physiology* **165**, 1255-1268 (2014).
36. De Giorgi J, *et al.* The Arabidopsis mature endosperm promotes seedling cuticle formation via release of sulfated peptides. *Developmental Cell* **56**, 3066-3081.e3065 (2021).

REVIEWER COMMENTS

Reviewer #1 (Remarks to the Author):

The manuscript has been thoroughly reviewed and authors replies as well as the new additions made to the work are satisfiable.

Reviewer #2 (Remarks to the Author):

The authors have addressed this reviewer's concerns and comments. The paper has been modified extensively, and several new experiments have been carried out. This has improved the manuscript and consolidated the observations made. The narrative is clearer in this new version.

I have a few comments for the authors to consider:

- Line 125 and following. Sup 2: I am not sure the paper needs a whistle-stop tour of all phenotypes unrelated to seeds' longevity. Because of space limitations, there is little space to explain the phenotypes looked at and non-specialist readers might struggle. It would be simpler to keep some of this for other papers. This section could be simplified, maybe keeping the no-PCD component of it.

- Sup Fig 5C: 'solubilization of MCA-II-a by various solvents/detergents'. Unless I misunderstand the experiment, I don't find it obvious that MCII-a is removed mainly by salt. At the labelled position on the western, it seems there is very little MCII-a in the various treatments and the Triton pellet has the most MCII-C. Not what I would expect. The protease assay is more convincing, and MCII-a seems to be on the outside of the membrane fraction.

- Line 205 The results of the split GFP experiment should be described better with one or 2 more sentences explaining the rationale of having GFP1-10 localised in various compartments. If short of space, the authors could remove the description of the split-GFP principle, which is well known and presented clearly in sup fig 7. i.e. remove 'This assay is based on GFP auto-assembly, when two polypeptides – the "detector" GFP1-10 (residues 1-214; fused to MCA-II-a) and the "tag" GFP11 (residues 215-230; fused to HDEL) – both non-fluorescing on their own, associate spontaneously and produce fluorescence'

- Ln 336 'we confirmed these results by specific immunostainings in root cells with a-CDC48/a-BIP2 ultracentrifuge fractionation analyses and immunostainings'. Please explain in the text what the analysis shows.

- Ln 356 'Indeed, the loss of function of MCA-II enhanced the sensitivity of seedlings to CB-5083'. To help the reader, Indicate briefly the phenotype e.g. enlarged root tips. Generally, this section about sensitivity to various drugs could use a few words to indicate the sensitivity phenotype.

- Ln 589 'seeds require an MCA-II-dependent proteostatic pathway (Fig. 5b, model)' . 5h, not 5b.

Reviewer #2 (Remarks to the Author):

The authors have addressed this reviewer's concerns and comments. The paper has been modified extensively, and several new experiments have been carried out. This has improved the manuscript and consolidated the observations made. The narrative is clearer in this new version. I have a few comments for the authors to consider:

- Line 125 and following. Sup 2: I am not sure the paper needs a whistle-stop tour of all phenotypes unrelated to seeds' longevity. Because of space limitations, there is little space to explain the phenotypes looked at and non-specialist readers might struggle. It would be simpler to keep some of this for other papers. This section could be simplified, maybe keeping the no-PCD component of it.

Thanks, we have contracted the discussion/results on the corresponding phenotype; please, see lines 126-136. We will use the removed dataset in another publication.

- Sup Fig 5C: 'solubilization of MCA-II-a by various solvents/detergents'. Unless I misunderstand the experiment, I don't find it obvious that MCII-a is removed mainly by salt. At the labelled position on the western, it seems there is very little MCII-a in the various treatments and the Triton pellet has the most MCII-C. Not what I would expect. The protease assay is more convincing, and MCII-a seems to be on the outside of the membrane fraction.

To avoid confusion due to the complicated nature of the interaction of MCAs with membranes, which also falls beyond our scope, we have removed this dataset.

- Line 205 The results of the split GFP experiment should be described better with one or 2 more sentences explaining the rationale of having GFP1-10 localised in various compartments. If short of space, the authors could remove the description of the split-GFP principle, which is well known and presented clearly in sup fig 7. i.e. remove 'This assay is based on GFP auto-assembly, when two polypeptides – the "detector" GFP1-10 (residues 1-214; fused to MCA-II-a) and the "tag" GFP11 (residues 215-230; fused to HDEL) – both non-fluorescing on their own, associate spontaneously and produce fluorescence'

We have now described better the corresponding assay, describing the rationale of having split GFP in various compartments, see lines 183-195. As it is fundamentally different from BiFC, to avoid any confusion among readers, we decided to keep part of the description for splitGFP rationale.

- Ln 336 'we confirmed these results by specific immunostainings in root cells with a-CDC48/a-BIP2 ultracentrifuge fractionation analyses and immunostainings'. Please explain in the text what the analysis shows.

Done (Lines 335-350)

- Ln 356 'Indeed, the loss of function of MCA-II enhanced the sensitivity of seedlings to CB-5083'. To help the reader, indicate briefly the phenotype e.g. enlarged root tips. Generally, this section about sensitivity to various drugs could use a few words to indicate the sensitivity phenotype.

Done, please see lines 348-365. We described the phenotypes and we include more context.

- Ln 589 'seeds require an MCA-II-dependent proteostatic pathway (Fig. 5b, model)' . 5h, not 5b.

Thanks for picking it up. Corrected.